# A Relative Error-Based Evaluation Framework of Heterogeneous Treatment Effect Estimators

**Jiayi Guo**[1]**, Haoxuan Li**[1]**, Ye Tian**[2]**, Peng Wu**[3]*
[1]Peking University, [2]Northeast Normal University, [3]Beijing Technology and Business University

## Abstract

While significant progress has been made in heterogeneous treatment effect (HTE) estimation, the evaluation of HTE estimators remains underdeveloped. In this article, we propose a robust evaluation framework based on relative error, which quantifies performance differences between two HTE estimators. We first derive the key theoretical conditions on the nuisance parameters that are necessary to achieve a robust estimator of relative error. Building on these conditions, we introduce novel loss functions and design a neural network architecture to estimate nuisance parameters and obtain robust estimation of relative error, thereby achieving reliable evaluation of HTE estimators. We provide large sample properties of the proposed relative error estimator. Furthermore, beyond evaluation, we propose a new learning algorithm for HTE that leverages both the previously HTE estimators and the nuisance parameters learned through our neural network architecture. Extensive experiments demonstrate that our evaluation framework supports reliable comparisons across HTE estimators, and the proposed learning algorithm for HTE exhibits desirable performance.

## 1 Introduction

The estimation of heterogeneous treatment effects (HTEs) has attracted substantial attention across a range of disciplines, including economics (Imbens & Rubin, 2015), marketing (Wager & Athey, 2018b), biology (Rosenbaum, 2020), and medicine (Hernán & Robins, 2020), due to its critical role in understanding individual-level treatment heterogeneity and supporting personalized, context-specific decision-making (Wu et al., 2024b;c). Various methods have been developed to estimate HTEs; see Kunzel et al. (2019); Caron et al. (2022) for comprehensive reviews. Despite their growing popularity, the evaluation of HTE estimators remain relatively underexplored (Gao, 2025). Assessing estimator performance is crucial in real-world applications, as a reliable evaluation framework can identify the most suitable methods (Curth & Van Der Schaar, 2023), directly impacting downstream tasks.

Evaluating HTEs is inherently challenging, as the ground truth is not available: only one potential outcome is observed for each individual, while HTEs are defined as the difference between two. To address this, researchers often rely on stringent model assumptions (Saito & YasuiAuthors, 2023; Mahajan et al., 2024; Wu & Wang, 2026) or preprocessing methods (e.g., matching, Rolling & Yang (2014)) to approximate the unobserved counterfactuals, and obtain an estimated treatment effect. Our work is motivated by Gao (2025), who introduced relative error to quantify the performance difference between two estimators, thereby reducing the bias caused by using inaccurately estimated treatment effects as ground truth.

Despite the significant contributions of Gao (2025), a notable limitation remains unaddressed. Their estimator requires that all nuisance estimators (propensity score and outcome models) are consistent at a rate faster than $n^{-1/4}$ to achieve consistency and valid confidence intervals for the relative error, which may be too stringent for real-world applications. In practice, the outcome models for potential outcomes heavily rely on model extrapolation. These models are trained separately within the treated and control groups, yet their predictions are applied across the entire dataset. When

---

*Corresponding author: pengwu@btbu.edu.cn.

there exists a significant distributional difference between the treated and control groups (Jeong & Namkoong, 2020; Jing Qin & Huang, 2024), the extrapolated predictions from these models are prone to inaccuracy, potentially leading to unreliable conclusions. Therefore, it is desirable to develop methods that reduce reliance on such extrapolation to ensure more robust and trustworthy evaluations.

To address this limitation, we propose a reliable evaluation approach for HTE estimation that retains the desirable properties of the method in Gao (2025), while relaxing the requirement for consistent outcome regression models. We show that the proposed estimator of relative error is $\sqrt{n}$-consistent, asymptotically normal, and yields valid confidence intervals, provided that the propensity score model is consistent at a rate faster than $n^{-1/4}$, even if the outcome regression model is inconsistent.

This robustness is achieved by carefully exploring the relationships between nuisance parameter models. We first derive the key conditions necessary for robustness and then design a novel loss function for estimating outcome regression models. Moreover, because our method still relies on a consistent propensity score model, we introduce new balance regularizers that reduce this dependence by encouraging the estimated scores to satisfy the balance property (Imai & Ratkovic, 2014), ensuring that inverse-propensity–weighted covariate functions have equal expectations across treatment groups.

Furthermore, by combining the novel loss function with balance regularizers, we design a new neural network architecture that more accurately estimates outcome regression and propensity score models, enabling more reliable relative error estimation and, in turn, more robust HTE evaluations. It should be mentioned that the proposed method does not require sample splitting and is numerically more tractable, which is discussed in Section 4.4. The main contributions are summarized as follows.

- We reveal the limitations of existing methods and, through theoretical analysis, derive key conditions for estimating the relative error that mitigate these limitations.

- We propose a reliable HTE evaluation method by designing novel loss functions and introducing a new neural network, enabling more robust estimation of relative error.

- We conduct extensive experiments to demonstrate the effectiveness of the proposed method.

## 2 PRELIMINARIES

### 2.1 PROBLEM SETTING

We introduce notations to formulate the problem of interest. For each individual $i$, let $A_i \in \mathcal{A} = \{0, 1\}$ denote the binary treatment variable, where $A_i = 1$ and $A_i = 0$ denote treatment and control. Let $X_i \in \mathcal{X} \subset \mathbb{R}^d$ be the pre-treatment covariates, and $Y_i \in \mathbb{R}$ be the outcome. We adopt the potential outcome framework in causal inference, defining $Y_i(0)$ and $Y_i(1)$ as the potential outcomes under $A_i = 0$ and $A_i = 1$, respectively. Since each individual receives either the treatment or the control, the observed outcome $Y_i$ satisfies $Y_i = A_i Y_i(1) + (1 - A_i) Y_i(0)$.

The individual treatment effect (ITE) is defined as $Y_i(1) - Y_i(0)$, which represents the treatment effect for a specific individual $i$. However, since only one of $(Y_i(0), Y_i(1))$ is observable, ITE is not identifiable without imposing strong assumptions (Pearl, 2009; Wu et al., 2025). In practice, the heterogeneous treatment effect (HTE) is often used to characterize "individual" treatment effects.

$$\tau(x) = \mathbb{E}[Y_i(1) - Y_i(0)|X_i = x],$$

which captures how treatment effects vary across individuals with different covariate values.

**Assumption 1** (Strongly Ignorability, (Rosenbaum & Rubin, 1983))**.** *(i)* $(Y_i(0), Y_i(1)) \perp\!\!\!\perp A_i \mid X_i$; *(ii)* $0 < e(x) \triangleq \mathbb{P}(A_i = 1 \mid X_i = x) < 1$ *for all* $x \in \mathcal{X}$, *where* $e(x)$ *is the propensity score.*

Under the standard strong ignorability assumption, HTE is identified as $\mu_1(x) - \mu_0(x)$ where $\mu_a(x) = \mathbb{E}[Y_i \mid X_i = x, A_i = a]$ for $a = 0, 1$ are the outcome regression functions, and various methods have been developed for estimating HTE (Wager & Athey, 2018a; Shalit et al., 2017a). Suppose we have a set of candidate HTE estimators trained on a training set, denoted by $\{\hat{\tau}_1(x), \cdots, \hat{\tau}_K(x)\}$. We aim to select the estimator with the highest accuracy of HTE using a test dataset $\{(X_i, A_i, Y_i), i = 1, \ldots, n\}$, which is of size $n$ and sampled from the super-population $\mathbb{P}$, and is independent of the training set.

## 2.2 EVALUATION METRICS: ABSOLUTE ERROR AND RELATIVE ERROR

For a given estimator $\hat{\tau}(x)$, its accuracy is typically evaluated using the MSE defined by

$$\phi(\hat{\tau}) \triangleq \mathbb{E}[(\hat{\tau}(X) - \tau(X))^2].$$

For any two estimators $\hat{\tau}_1(x)$ and $\hat{\tau}_2(x)$, the difference in their MSE is

$$\delta(\hat{\tau}_1, \hat{\tau}_2) \triangleq \phi(\hat{\tau}_1) - \phi(\hat{\tau}_2) = \mathbb{E}[\hat{\tau}_1^2(X) - \hat{\tau}_2^2(X) - 2(\hat{\tau}_1(X) - \hat{\tau}_2(X))\tau(X)].$$

Gao (2025) refers to $\phi(\hat{\tau})$ and $\delta(\hat{\tau}_1, \hat{\tau}_2)$ as the absolute error and relative error, respectively. In practice, absolute error is used much more frequently than relative error. However, Gao (2025) established, through theoretical analysis and experiments, that relative error is preferable to absolute error; details are provided in Section 3. Intuitively, one can see that the key advantage of using relative error over absolute error lies in that it only relies on the first-order term of the unobserved $\tau$, which reduces the impact of estimation error in $\tau$.

Several studies (Gutierrez & Gérardy, 2017; Powers et al., 2017) have used $\mathbb{E}[(Y(1)-Y(0)-\hat{\tau}(X))^2]$ to evaluate the estimator $\hat{\tau}(x)$. However, its estimator requires knowing the values of $(Y(0), Y(1))$ that are not observable in real-world applications. We note that

$$\mathbb{E}[(Y(1) - Y(0) - \hat{\tau}(X))^2] = \mathbb{E}[(\hat{\tau}(X) - \tau(X))^2 + \mathbb{E}[\text{Var}(Y(1) - Y(0)|X)],$$

where the second term on the right-hand side is independent of $\hat{\tau}(x)$. Thus, this metric is essentially equivalent to the absolute error $\phi(\hat{\tau})$, and we will not discuss it further. For clarity, we provide a notation summary tab, but due to limited space, we present it in Appendix A.

## 3 MOTIVATION

In this section, we briefly discuss the advantages of using relative error over absolute error, and then analyze the limitations of the method in Gao (2025), which motivate this work.

The key theoretical advantage of relative error over absolute error is demonstrated through its semiparametric efficient estimators. A semiparametric efficient estimator is considered optimal (or gold standard) in the sense that it has the smallest asymptotic variance under regularity conditions (Newey, 1990; van der Vaart, 1998) given the observed test data. Let $\{\tilde{e}(x), \tilde{\mu}_1(x), \tilde{\mu}_0(x)\}$ be the estimators of $\{e(x), \mu_1(x), \mu_0(x)\}$, which are the nuisance functions to construct semiparametric efficient estimators of absolute error and relative error. Denote $\tilde{\tau}(x) = \tilde{\mu}_1(x) - \tilde{\mu}_0(x)$.

**Absolute Error.** Given $\hat{\tau}(x)$, an estimator of $\phi(\hat{\tau})$ is constructed as

$$\hat{\phi}(\hat{\tau}) = \frac{1}{n}\sum_{i=1}^{n}\{\tilde{\tau}(X_i) - \hat{\tau}(X_i)\}^2 + 2(\tilde{\tau}(X_i) - \hat{\tau}(X_i))\left(\frac{A_i(Y_i - \tilde{\mu}_1(X_i))}{\tilde{e}(X_i)} - \frac{(1 - A_i)(Y_i - \tilde{\mu}_0(X_i))}{1 - \tilde{e}(X_i)}\right).$$

Under Assumption 1, $\hat{\phi}(\hat{\tau})$ is $\sqrt{n}$-consistent, asymptotically normal, and semiparametric efficient, provided that the estimated nuisance parameter satisfy the key Condition 1.

**Condition 1.** $\mathbb{E}[(\tilde{e}(X) - e(X))^2] = o_{\mathbb{P}}(n^{-1/2})$, $\mathbb{E}[(\tilde{\mu}_a(X) - \mu_a(X))^2] = o_{\mathbb{P}}(n^{-1/2})$ *for* $a = 0, 1$.

**Relative Error.** Likewise, we can construct the estimator of $\delta(\hat{\tau}_1, \hat{\tau}_2)$ given as

$$\hat{\delta}(\hat{\tau}_1, \hat{\tau}_2) = \frac{1}{n}\sum_{i=1}^{n}\varphi(Z_i; \tilde{\mu}_0, \tilde{\mu}_1, \tilde{e}), \text{ where } Z_i \triangleq (A_i, X_i, Y_i),$$

$$\varphi(Z_i; \tilde{\mu}_0, \tilde{\mu}_1, \tilde{e}) \triangleq \{\hat{\tau}_1^2(X_i) - \hat{\tau}_2^2(X_i)\} - 2(\hat{\tau}_1(X_i) - \hat{\tau}_2(X_i))\cdot$$
$$\left(\frac{A_i(Y_i - \tilde{\mu}_1(X_i))}{\tilde{e}(X_i)} + \tilde{\mu}_1(X_i) - \frac{(1 - A_i)(Y_i - \tilde{\mu}_0(X_i))}{1 - \tilde{e}(X_i)} - \tilde{\mu}_0(X_i)\right).$$

Under Assumption 1, the $\sqrt{n}$-consistency, asymptotic normality, and semiparametric efficiency of $\hat{\delta}(\hat{\tau}_1, \tau_2)$ rely on the key Condition 2 below.

**Condition 2.** $\mathbb{E}[|\tilde{\mu}_a(X) - \mu_a(X)||\tilde{e}(X) - e(X)|] = o_{\mathbb{P}}(n^{-1/2})$.

Condition 2 is strictly weaker than Condition 1. Moreover, the estimator $\hat{\delta}(\hat{\tau}_1, \hat{\tau}_2)$ offers several additional advantages over $\hat{\phi}(\hat{\tau})$, see Appendix B for more details.

**Motivation.** Although $\hat{\delta}(\hat{\tau}_1, \hat{\tau}_2)$ has several desirable properties, a notable limitation is that Condition 2 requires all nuisance parameter estimators to be consistent (as $\tilde{e}(x)$ and $\tilde{\mu}_a(x)$ generally converge at most at the rate of $n^{-1/2}$), which may be overly restrictive in practical settings. In practice, the outcome regression model $\tilde{\mu}_a(x)$ is learned from the data with $A = a$ and then applied to the entire data. It heavily relies on model extrapolation, as there is often a significant distributional difference between the data with $A = a$ and $A = 1 - a$ (Jeong & Namkoong, 2020; Jing Qin & Huang, 2024). As a result, $\tilde{\mu}_a(x)$ is likely to be inaccurate and biased, violating Assumption 2. Therefore, it is beneficial and practical to develop methods that rely less on model extrapolation. In contrast, estimating the propensity score does not involve any model extrapolation, as the score is learned from the full dataset. As a result, the model is not required to predict outside the support of the observed covariates, making propensity score estimation substantially less susceptible to extrapolation errors and more robust to flexible machine learning specifications.

A natural and practical question arises: Can we develop a method for estimating $\delta(\hat{\tau}_1, \hat{\tau}_2)$, that retains all the desirable properties of $\hat{\delta}(\hat{\tau}_1, \hat{\tau}_2)$, while allowing for bias in $\tilde{\mu}_a(x)$ (relaxing Condition 2)? In this article, we show that this is achievable by carefully exploiting the connection between the propensity score and outcome regression models, and by designing appropriate loss functions.

## 4  PROPOSED METHOD

In this section, we propose a novel method for estimating the relative error $\delta(\hat{\tau}_1, \hat{\tau}_2)$ that retains the desirable properties of $\hat{\delta}(\hat{\tau}_1, \hat{\tau}_2)$ while simultaneously being robust to bias in $\tilde{\mu}_a(x)$ for $a = 0, 1$. We consider the following working models for the propensity score and outcome regression functions,

$$e(X) = \mathbb{P}(A = 1 \mid X) = e(\Phi(X), \gamma) = \frac{\exp(\Phi(X)^\intercal \gamma)}{1 + \exp(\Phi(X)^\intercal \gamma)}, \tag{1}$$

$$\mu_a(X) = \mathbb{E}(Y \mid X, A = a) = \mu_a(\Phi(X), \beta_a) = \Phi(X)^\intercal \beta_a, \quad a = 0, 1, \tag{2}$$

where $\Phi(X)$ is a representation of $X$ adaptively learned from data. It facilitates theoretical analysis of the connections between models and helps establish more desirable theoretical properties and is widely used in the literature (e.g. (Shi et al., 2019b)) and usually yield more stable numerical results. That said, to illustrate the role of representation sharing more concretely, we also report in Appendix F.2 the empirical performance of the method when no shared representation is used.

To quantify the bias of $\tilde{\mu}_a(x)$, it is crucial to distinguish between the working model and the true model. We say a working model is misspecified if the true model does not belong to the working model class, and it is correctly specified if the true model is within the working model class. For clarity, we provide a misspecified example in Appendix B.

For models (1) and (2), let $\tilde{\gamma}$ and $\check{\beta}_a$ denote the estimators of $\gamma$ and $\beta_a$, respectively. Define $\bar{\gamma}$ and $\bar{\beta}_a$ as the probability limits of $\tilde{\gamma}$ and $\check{\beta}_a$, and denote $\bar{e}(X) = e(\Phi(X), \bar{\gamma})$ and $\bar{\mu}_a(X) = \mu_a(\Phi(X), \bar{\beta}_a)$. If model (1) is specified correctly, $e(X) = \bar{e}(X)$; otherwise, $e(X) \neq \bar{e}(X)$ and their difference represents the systematic bias induced by model misspecification. Similarly, if model (2) is correctly specified, $\bar{\mu}_a(X) = \mu_a(X)$; otherwise, $\bar{\mu}_a(X) \neq \mu_a(X)$. *It is important to note that $(\tilde{\gamma}, \check{\beta}_0, \check{\beta}_1)$ always converges to $(\bar{\gamma}, \bar{\beta}_0, \bar{\beta}_1)$, regardless of whether models (1) and (2) are correctly specified.*

### 4.1  BASIC IDEA

Before delving into the details, we outline its basic idea to provide an intuitive understanding.

**First**, to retain the semiparametric efficiency, the proposed estimator $\delta(\hat{\tau}_1, \hat{\tau}_2)$ preserves the same form of $\hat{\delta}(\hat{\tau}_1, \tau_2)$, which is given as

$$\check{\delta}(\hat{\tau}_1, \hat{\tau}_2; \tilde{\gamma}, \check{\beta}_0, \check{\beta}_1) = \frac{1}{n} \sum_{i=1}^{n} \varphi(Z_i; \check{\mu}_0, \check{\mu}_1, \check{e}),$$

where $\check{e}(X) = e(\Phi(X), \tilde{\gamma})$ and $\check{\mu}_a(X) = \mu_a(\Phi(X), \check{\beta}_a)$ for $a = 0, 1$. Although $\check{\delta}(\hat{\tau}_1, \hat{\tau}_2; \tilde{\gamma}, \check{\beta}_0, \check{\beta}_1)$ shares the same algebraic form as $\delta(\hat{\tau}_1, \hat{\tau}_2)$, its construction is fundamentally different, which

makes the resulting estimator substantially more robust to bias in $\check{\mu}_a(X)$. This distinction is a key methodological contribution of our work.

**Second**, we analyze the key conditions necessary to achieve robustness to biases in $\check{\mu}_a(X)$. By a Taylor expansion of $\check{\delta}(\hat{\tau}_1, \hat{\tau}_2; \check{\gamma}, \check{\beta}_0, \check{\beta}_1)$ with respect to $(\bar{\gamma}, \bar{\beta}_0, \bar{\beta}_1)$, we have that

$$\check{\delta}(\hat{\tau}_1, \hat{\tau}_2; \check{\gamma}, \check{\beta}_0, \check{\beta}_1) - \check{\delta}(\hat{\tau}_1, \hat{\tau}_2; \bar{\gamma}, \bar{\beta}_0, \bar{\beta}_1) = \Delta_\gamma^\intercal(\check{\gamma} - \bar{\gamma}) + \Delta_{\beta_0}^\intercal(\check{\beta}_0 - \bar{\beta}_0) + \Delta_{\beta_1}^\intercal(\check{\beta}_1 - \bar{\beta}_1)$$
$$+ O_\mathbb{P}((\check{\gamma} - \bar{\gamma})^2 + (\check{\gamma} - \bar{\gamma})(\check{\beta}_1 - \bar{\beta}_1) + (\check{\gamma} - \bar{\gamma})(\check{\beta}_0 - \bar{\beta}_0)),$$

where

$$\Delta_\gamma = -\frac{1}{n} \sum_{i=1}^n 2(\hat{\tau}_1(X_i) - \hat{\tau}_2(X_i)) \left( \frac{A_i(1 - \bar{e}(X_i))(Y_i - \bar{\mu}_1(X_i))}{\bar{e}(X_i)} + \frac{(1 - A_i)\bar{e}(X_i)(Y_i - \bar{\mu}_0(X_i))}{1 - \bar{e}(X_i)} \right) \Phi(X_i),$$

$$\Delta_{\beta_0} = -\frac{1}{n} \sum_{i=1}^n 2(\hat{\tau}_1(X_i) - \hat{\tau}_2(X_i)) \left( 1 - \frac{1 - A_i}{1 - \bar{e}(X_i)} \right) \Phi(X_i),$$

$$\Delta_{\beta_1} = \frac{1}{n} \sum_{i=1}^n 2(\hat{\tau}_1(X_i) - \hat{\tau}_2(X_i)) \left( 1 - \frac{A_i}{\bar{e}(X_i)} \right) \Phi(X_i).$$

Under mild conditions (see Theorem 1), the last term of above Taylor expansion is $o_\mathbb{P}(n^{-1/2})$. We note that $\check{\delta}(\hat{\tau}_1, \hat{\tau}_2; \bar{\gamma}, \bar{\beta}_0, \bar{\beta}_1)$ is a $\sqrt{n}$-consistent and asymptotically normal estimator of $\delta(\hat{\tau}_1, \hat{\tau}_2)$ if either $\check{e}(x)$ is correctly specified, or $(\check{\mu}_0(x), \check{\mu}_1(x))$ is correctly specified. Thus, it is robust to biases in $\check{\mu}_a(x)$ for $a = 0, 1$ and is the ideal estimator we aim to obtain. To ensure that $\check{\delta}(\hat{\tau}_1, \hat{\tau}_2; \check{\gamma}, \check{\beta}_0, \check{\beta}_1)$ has the same asymptotic properties as $\check{\delta}(\hat{\tau}_1, \hat{\tau}_2; \bar{\gamma}, \bar{\beta}_0, \bar{\beta}_1)$, we require that

$$\Delta_\gamma^\intercal(\check{\gamma} - \bar{\gamma}) + \Delta_{\beta_0}^\intercal(\check{\beta}_0 - \bar{\beta}_0) + \Delta_{\beta_1}^\intercal(\check{\beta}_1 - \bar{\beta}_1) = o_\mathbb{P}(n^{-1/2}), \tag{3}$$

*even when* $(\check{\mu}_0(x), \check{\mu}_1(x))$ *is misspecified.* Note that $(\check{\gamma}, \check{\beta}_0, \check{\beta}_1)$ always converges to $(\bar{\gamma}, \bar{\beta}_0, \bar{\beta}_1)$. To satisfy Eq. (3), it suffices for $\Delta_\gamma$, $\Delta_{\beta_0}$, and $\Delta_{\beta_1}$ to converge to zero at a certain rate. By the central limit theorem, $\Delta_\gamma - \mathbb{E}[\Delta_\gamma] = O_\mathbb{P}(n^{-1/2})$, $\Delta_{\beta_0} - \mathbb{E}[\Delta_{\beta_0}] = O_\mathbb{P}(n^{-1/2})$, and $\Delta_{\beta_1} - \mathbb{E}[\Delta_{\beta_1}] = O_\mathbb{P}(n^{-1/2})$. Thus, Eq. (3) holds provided that

$$\mathbb{E}[\Delta_\gamma] = 0, \ \mathbb{E}[\Delta_{\beta_0}] = 0, \ \mathbb{E}[\Delta_{\beta_1}] = 0,$$

which is equivalent to the following equations:

$$\begin{cases} \mathbb{E}\left[ (\hat{\tau}_1(X_i) - \hat{\tau}_2(X_i)) \left( \frac{A_i(1 - \bar{e}(X_i))(Y_i - \bar{\mu}_1(X_i))}{\bar{e}(X_i)} + \frac{(1 - A_i)\bar{e}(X_i)(Y_i - \bar{\mu}_0(X_i))}{1 - \bar{e}(X_i)} \right) \Phi(X_i) \right] = 0, \\ \mathbb{E}\left[ (\hat{\tau}_1(X_i) - \hat{\tau}_2(X_i)) \left( 1 - \frac{A_i}{\bar{e}(X_i)} \right) \Phi(X_i) \right] = 0, \\ \mathbb{E}\left[ (\hat{\tau}_1(X_i) - \hat{\tau}_2(X_i)) \left( 1 - \frac{1 - A_i}{1 - \bar{e}(X_i)} \right) \Phi(X_i) \right] = 0. \end{cases} \tag{4}$$

### 4.2 NOVEL LOSS FOR NUISANCE PARAMETER ESTIMATION

To ensure that the first term in Eq. (4) holds, we design the weighted least square loss function for $(\beta_0, \beta_1)$ as follows:

$$\mathcal{L}_{\text{wls}}(\beta_0, \beta_1; \check{\gamma}) = \frac{1}{n} \sum_{i=1}^n (\hat{\tau}_1(X_i) - \hat{\tau}_2(X_i)) \left[ \frac{(1 - A_i)\check{e}(X_i)\{Y_i - \Phi(X)^\intercal\beta_0\}^2}{1 - \check{e}(X_i)} + \frac{A_i(1 - \hat{e}(X_i))\{Y_i - \Phi(X)^\intercal\beta_1\}^2}{\hat{e}(X_i)} \right].$$

These loss functions imply that $(\bar{\beta}_0, \bar{\beta}_1) \triangleq \arg\min_{\beta_a} \mathbb{E}[\mathcal{L}_{\text{wls}}(\beta_0, \beta_1; \bar{\gamma})]$. By setting $\partial\mathbb{E}[\mathcal{L}_{\text{wls}}(\beta_0, \beta_1; \bar{\gamma})]/\partial\beta_0\big|_{\beta_0 = \bar{\beta}_0} = 0$ and $\partial\mathbb{E}[\mathcal{L}_{\text{wls}}(\beta_0, \beta_1; \bar{\gamma})]/\partial\beta_1\big|_{\beta_1 = \bar{\beta}_1} = 0$, one can see that the first term in Eq. (4) holds even if $(\check{\mu}_0(x), \check{\mu}_1(x))$ is misspecified.

For learning $\gamma$, observe that Eq. (4) specifies $2d$ linear constraints, whereas $\gamma \in \mathbb{R}^d$ contains only $d$ free parameters, making the system inherently over-constrained. To obtain a feasible solution, we adopt a soft relaxation strategy inspired by the soft-margin formulation of support vector machines (Murphy, 2022): we introduce slack variables $\xi, \eta \in \mathbb{R}^d$ that permit controlled deviations

from the constraints, and penalize their magnitudes in the optimization objective. Formally, we solve:

$$
\min_{\gamma, \xi, \eta} \quad -\frac{1}{n} \sum_{i=1}^{n} \left[ A_i \log(e(X_i)) + (1 - A_i) \log(1 - e(X_i)) \right] + c \sum_{j=1}^{d} (\xi_j + \eta_j)
$$

$$
\text{s.t.} \quad e(X_i) = \frac{\exp(\Phi(X_i)^\top \gamma)}{1 + \exp(\Phi(X_i)^\top \gamma)}, \quad i = 1, \dots, n,
$$

$$
\left| \frac{1}{n} \sum_{i=1}^{n} (\hat{\tau}_1(X_i) - \hat{\tau}_2(X_i)) \left(1 - \frac{A_i}{e(X_i)}\right) \Phi_j(X_i) \right| \leq \xi_j, \quad \forall j,
$$

$$
\left| \frac{1}{n} \sum_{i=1}^{n} (\hat{\tau}_1(X_i) - \hat{\tau}_2(X_i)) \left(1 - \frac{1 - A_i}{1 - e(X_i)}\right) \Phi_j(X_i) \right| \leq \eta_j, \quad \forall j,
$$

$$
\xi_j, \eta_j \geq 0, \quad j = 1, \dots, d.
$$

where $c$ is a given hyperparameter. In practice, we convert the above constrained optimization into two unconstrained loss terms, including the cross-entropy loss

$$
\mathcal{L}_{\text{ce}} = -\frac{1}{n} \sum_{i=1}^{n} \left[ A_i \log(e(X_i)) + (1 - A_i) \log(1 - e(X_i)) \right],
$$

and the balance regularizer

$$
\mathcal{L}_{\text{const}} = c \sum_{j=1}^{d} (\xi_j + \eta_j) + \rho \cdot \left\| \begin{bmatrix} \max\left\{ \left| \frac{1}{n} \sum_{i=1}^{n} (\hat{\tau}_1(X_i) - \hat{\tau}_2(X_i)) \left(1 - \frac{A_i}{e(X_i)}\right) \Phi(X_i) \right| - \xi, \, 0 \right\} \\ \max\left\{ \left| \frac{1}{n} \sum_{i=1}^{n} (\hat{\tau}_1(X_i) - \hat{\tau}_2(X_i)) \left(1 - \frac{1 - A_i}{1 - e(X_i)}\right) \Phi(X_i) \right| - \eta, \, 0 \right\} \\ \max(-\xi, \, 0) \\ \max(-\eta, \, 0) \end{bmatrix} \right\|_2,
$$

where $\rho > 0$ is a penalty parameter encouraging the satisfaction of the original constraints. As we show in Appendix F.4, this relaxation is effective in practice, and the resulting unconstrained formulation still enforces the original conditions to a high degree of accuracy.

### 4.3 CONSTRUCTING NEURAL NETWORK

Building on the constraint loss from Section 4.2, we adopt a neural architecture derived from the Dragonnet framework (Shi et al., 2019a). The proposed network takes input features $x \in \mathbb{R}^d$, and first passes them through multiple fully connected layers to produce the shared representation $\Phi(x) \in \mathbb{R}^m$. This representation is then fed into three separate heads: a control outcome head $\mu_0(x)$, predicting the potential outcome under control; a treated outcome head $\mu_1(x)$, predicting the potential outcome under treatment; a treatment head $e(x)$, estimating the propensity score via a sigmoid activation.

The control outcome head and the treated outcome head contribute to the weighted least square loss $\mathcal{L}_{\text{wls}}$, while $\mathcal{L}_{\text{ce}}$ and $\mathcal{L}_{\text{const}}$ are computed by the treatment head and the shared representation. During training, we minimize the total training loss given by:

$$
\mathcal{L} = \mathcal{L}_{\text{wls}} + \lambda_1 \mathcal{L}_{\text{ce}} + \lambda_2 \mathcal{L}_{\text{const}}.
$$

This formulation encourages the propensity model $e(X)$ and the outcome model $\mu_a(X)$ to satisfy Eq. (4), providing a reliable estimation that can be used in computing the estimated relative error $\hat{\delta}$ mentioned in Section 3. For clarity, we provide a schematic illustration of the network architecture in Appendix D.

### 4.4 THEORETICAL ANALYSIS

We analyze the large sample properties of the proposed estimator $\check{\delta}(\hat{\tau}_1, \hat{\tau}_2; \check{\gamma}, \check{\beta}_0, \check{\beta}_1)$.

**Theorem 1.** *If the propensity score model is correctly specified, and $\check{\gamma}$, $\check{\beta}_0$ as well as $\check{\beta}_1$ converge to their probability limits at a rate faster than $n^{-1/4}$, then we have*

$$
\sqrt{n}\{\check{\delta}(\hat{\tau}_1, \hat{\tau}_2; \check{\gamma}, \check{\beta}_0, \check{\beta}_1) - \delta(\hat{\tau}_1, \hat{\tau}_2)\} \xrightarrow{d} \mathcal{N}(0, \sigma^2),
$$

*where $\sigma^2 = \text{Var}\{\varphi(Z; \bar{u}_0, \bar{u}_1, \bar{e})\}$ and $\xrightarrow{d}$ means convergence in distribution.*

Theorem 1 shows that the proposed estimator is $\sqrt{n}$-consistent and asymptotically normal. These properties hold even when the outcome regression model is misspecified, as long as $\check{\gamma}$, $\check{\beta}_0$, and $\check{\beta}_1$ converge to their respective probability limits at a rate faster than $n^{-1/4}$. This condition is readily satisfied, as $(\check{\gamma}, \check{\beta}_0, \check{\beta}_1)$ always converge to their probability limits $(\bar{\gamma}, \bar{\beta}_0, \bar{\beta}_1)$, and a variety of flexible machine learning methods can achieve the required convergence rates (Chernozhukov et al., 2018; Semenova & Chernozhukov, 2021).

Based on Theorem 1, we can obtain a valid asymptotic $(1 - \eta)$ confidence interval of $\delta(\hat{\tau}_1, \hat{\tau}_2)$.

**Proposition 2.** *Under the conditions in Theorem 1, a consistent estimator of $\sigma^2$ is*

$$\hat{\sigma}^2 = \frac{1}{n} \sum_{i=1}^{n} \left\{ \varphi(Z_i; \check{u}_0, \check{u}_1, \check{e}) - \check{\delta}(\hat{\tau}_1, \hat{\tau}_2; \check{\gamma}, \check{\beta}_0, \check{\beta}_1) \right\}^2,$$

*an asymptotic $(1 - \eta)$ confidence interval for $\delta(\hat{\tau}_1, \hat{\tau}_2)$ is $\check{\delta}(\hat{\tau}_1, \hat{\tau}_2; \check{\gamma}, \check{\beta}_0, \check{\beta}_1) \pm z_{\eta/2} \sqrt{\hat{\sigma}^2/n}$, where $z_{\eta/2}$ is the $(1 - \eta/2)$ quantile of the standard normal distribution.*

Proposition 2 shows that a valid asymptotic confidence interval for $\delta(\hat{\tau}_1, \hat{\tau}_2)$ is achievable even with a misspecified outcome model, unlike previous methods that require correct specification. This further indicates the robustness of the proposed method. We also emphasize that, unlike (Gao, 2025), our proposed methodology does not require sample splitting. The key derivation in Section 4.1, as well as the proofs of Theorem 1 and Proposition 2 in this section, are conducted using the full dataset without sample splitting.

Besides, we clarify that the correct specification of propensity score in Theorem 1 and Proposition 2 is a mild condition. First, as $\Phi(X)$ can be adaptively learned from the data, we are likely to gain the true working model using flexible neural networks. Despite this, we provide a sensitivity analysis on the correct specification on $\Phi(X)$ (Appendix F.3). In addition, we could adjust the model specification iteratively by balancing checking as follows: (1) Model specification and estimation: specify the model and estimate the propensity scores; (2) Covariates balance checking (Imai & Ratkovic, 2014). If the covariates balance is inadequate, adjust the model specification and re-estimate the propensity scores; if the balance is satisfactory, proceed with the estimated propensity scores.

## 5 ENHANCED ESTIMATION OF HETEROGENEOUS TREATMENT EFFECTS

In this section, building on the evaluation framework proposed in Section 4, we extend the idea to develop a learning method for HTE. In general, a reliable evaluation method can naturally serve as a basis for developing a learning method. In our proposed approach, for any given pair of HTE estimators $\hat{\tau}_k(x)$ and $\hat{\tau}_{k'}(x)$, the proposed neural network architecture introduced in Section 4.3 can output the corresponding estimates of outcome regression functions. We denote them as $\check{\mu}_0(x; \hat{\tau}_k, \hat{\tau}_{k'})$ and $\check{\mu}_1(x; \hat{\tau}_k, \hat{\tau}_{k'})$, emphasizing their dependence on $\hat{\tau}_k(x)$ and $\hat{\tau}_{k'}(x)$. This leads to a new HTE estimator, defined as

$$\check{\tau}(x; \hat{\tau}_k, \hat{\tau}_{k'}) = \check{\mu}_1(x; \hat{\tau}_k, \hat{\tau}_{k'}) - \check{\mu}_0(x; \hat{\tau}_k, \hat{\tau}_{k'}).$$

Clearly, the performance of the estimator heavily depends on the choice of HTE estimators $\check{\tau}(x; \hat{\tau}_k, \hat{\tau}_{k'})$. However, due to the fundamental challenge in evaluating HTE (i.e., the absence of ground truth), it is difficult to develop a direct strategy for selecting them. To mitigate this issue, we propose the following aggregation strategy for estimating HTE,

$$\check{\tau}(x) = \frac{2}{|\mathcal{K}|(|\mathcal{K}| - 1)} \sum_{k, k' \in \mathcal{K}} \check{\mu}_1(x; \hat{\tau}_k, \hat{\tau}_{k'}) - \check{\mu}_0(x; \hat{\tau}_k, \hat{\tau}_{k'})$$

where $\mathcal{K} = \{1, 2, \ldots, K\}$ is the index set for the candidate HTE estimators. This aggregated estimator aims to stabilize and improve the estimation of HTE by averaging over all pairs of candidate estimators. When $K$ is large, averaging over all pairs can be computationally burdensome. In such cases, one can randomly select a subset of pairs $\check{\tau}(x; \hat{\tau}_k, \hat{\tau}_{k'})$ and compute their average instead. Surprisingly, our experiments show that this estimator performs exceptionally well, even surpassing the performance of any single candidate estimator.

Figure 1: Coverage rate on IHDP and Twins.

Figure 2: Selection accuracy on IHDP and Twins.

Table 1: HTE estimation performance on the IHDP and Twins datasets (in-sample and out-of-sample). The best results are bolded.

| Method | IHDP | | | | Twins | | | |
|---|---|---|---|---|---|---|---|---|
| | $\sqrt{\epsilon_{\text{PEHE}}^{\text{in}}}$ | $\epsilon_{\text{ATE}}^{\text{in}}$ | $\sqrt{\epsilon_{\text{PEHE}}^{\text{out}}}$ | $\epsilon_{\text{ATE}}^{\text{out}}$ | $\sqrt{\epsilon_{\text{PEHE}}^{\text{in}}}$ | $\epsilon_{\text{ATE}}^{\text{in}}$ | $\sqrt{\epsilon_{\text{PEHE}}^{\text{out}}}$ | $\epsilon_{\text{ATE}}^{\text{out}}$ |
| LinDML | $1.053 \pm 0.134$ | $0.580 \pm 0.152$ | $1.085 \pm 0.187$ | $0.574 \pm 0.176$ | $0.295 \pm 0.005$ | $0.013 \pm 0.009$ | $0.296 \pm 0.008$ | $0.013 \pm 0.010$ |
| SpaDML | $0.832 \pm 0.119$ | $0.252 \pm 0.185$ | $0.866 \pm 0.112$ | $0.280 \pm 0.183$ | $0.300 \pm 0.008$ | $0.046 \pm 0.030$ | $0.303 \pm 0.010$ | $0.046 \pm 0.033$ |
| CForest | $0.891 \pm 0.121$ | $0.419 \pm 0.182$ | $0.903 \pm 0.127$ | $0.403 \pm 0.185$ | $0.297 \pm 0.005$ | $0.012 \pm 0.008$ | $0.306 \pm 0.008$ | $0.013 \pm 0.011$ |
| X-Learner | $0.971 \pm 0.178$ | $0.196 \pm 0.137$ | $0.987 \pm 0.196$ | $0.207 \pm 0.141$ | $0.293 \pm 0.005$ | $0.022 \pm 0.014$ | $0.294 \pm 0.008$ | $0.024 \pm 0.016$ |
| S-Learner | $0.920 \pm 0.102$ | $0.212 \pm 0.100$ | $0.950 \pm 0.111$ | $0.205 \pm 0.117$ | $0.298 \pm 0.011$ | $0.057 \pm 0.042$ | $0.299 \pm 0.010$ | $0.059 \pm 0.042$ |
| TARNet | $0.896 \pm 0.054$ | $0.279 \pm 0.084$ | $0.920 \pm 0.070$ | $0.266 \pm 0.117$ | $0.292 \pm 0.011$ | $0.090 \pm 0.047$ | $0.294 \pm 0.019$ | $0.091 \pm 0.045$ |
| Dragonnet | $0.840 \pm 0.046$ | $0.124 \pm 0.089$ | $0.867 \pm 0.087$ | $0.134 \pm 0.092$ | $0.292 \pm 0.004$ | $0.080 \pm 0.008$ | $0.290 \pm 0.007$ | $0.092 \pm 0.011$ |
| DRCFR | $0.741 \pm 0.068$ | $0.186 \pm 0.138$ | $0.760 \pm 0.090$ | $0.185 \pm 0.135$ | $0.290 \pm 0.004$ | $0.075 \pm 0.007$ | $0.288 \pm 0.007$ | $0.076 \pm 0.010$ |
| SCIGAN | $0.898 \pm 0.374$ | $0.358 \pm 0.509$ | $0.919 \pm 0.369$ | $0.358 \pm 0.502$ | $0.296 \pm 0.037$ | $0.041 \pm 0.044$ | $0.293 \pm 0.039$ | $0.040 \pm 0.047$ |
| DESCN | $0.793 \pm 0.187$ | $0.133 \pm 0.106$ | $0.835 \pm 0.197$ | $0.140 \pm 0.112$ | $0.296 \pm 0.060$ | $0.059 \pm 0.043$ | $0.293 \pm 0.063$ | $0.058 \pm 0.042$ |
| ESCFR | $0.802 \pm 0.041$ | $0.111 \pm 0.070$ | $0.841 \pm 0.074$ | $0.135 \pm 0.076$ | $0.290 \pm 0.004$ | $0.075 \pm 0.007$ | $0.288 \pm 0.007$ | $0.076 \pm 0.010$ |
| Ours | $\mathbf{0.638 \pm 0.138}$ | $\mathbf{0.090 \pm 0.087}$ | $\mathbf{0.670 \pm 0.150}$ | $\mathbf{0.105 \pm 0.099}$ | $\mathbf{0.284 \pm 0.005}$ | $\mathbf{0.009 \pm 0.005}$ | $\mathbf{0.286 \pm 0.007}$ | $\mathbf{0.009 \pm 0.006}$ |

# 6 EXPERIMENTS

## 6.1 EXPERIMENTAL SETUP

**Datasets and Processing.** Following previous studies (Yoon et al., 2018; Yao et al., 2018; Louizos et al., 2017), we choose one semi-synthetic dataset **IHDP**, and two real datasets, **Twins** and **Jobs**, to conduct our experiments. The **Twins** dataset is constructed from all twin births in the United States between 1989 and 1991 (Almond et al., 2005), owning 5271 samples with 28 different covariates. The **IHDP** dataset is used to estimate the effect of specialist home visits on infants' future cognitive test scores, containing 747 samples (139 treated and 608 control), each with 25 pre-treatment covariates, while the **Jobs** dataset focuses on estimating the impact of job training programs on individuals' employment status, including 297 treated units, 425 control units from the experimental sample, and 2490 control units from the observational sample. We provide more dataset details in the Appendix F.1. We randomly split each dataset into training and test sets in a 2:1 ratio, and repeat the experiments 50 times for the **Twins**, 100 times for the **IHDP**, and 20 times for the **Jobs**.

**Evaluation Metrics.** We consider two classes of evaluation metrics below.

- We assess the proposed relative error estimator using two key metrics: (i) the coverage probability of its confidence interval (named coverage rate), and (ii) the probability of correctly identifying the better estimator (i.e., selecting the true winner, named selection accuracy). In practice, we only pick the winner when the confidence interval for the relative error does not contain zero, otherwise, no selection will be made. We calculate the coverage rate of the **targeted 90%** confidence intervals.

- For evaluating the performance of HTE estimation of our novel network, following previous studies (Shalit et al., 2017a; Shi et al., 2019b; Louizos et al., 2017), we compute the Precision in Estimation of Heterogeneous Effect (PEHE) (and, 2011), where $\sqrt{\epsilon_{\text{PEHE}}} = \sqrt{\frac{1}{n}\sum_{i=1}^{n}\left(\hat{\tau}(x_i) - \tau(x_i)\right)^2}$, and the absolute error on the ATE, $\epsilon_{\text{ATE}} = |\text{ATE} - \widehat{\text{ATE}}|$, where $\text{ATE} = \frac{1}{n}\sum_{i=1}^{n}(y_i^1 - y_j^0)$, in which $y_i^1$ and $y_j^0$ are the true potential outcomes.

**Baselines and Experimental Details.** To evaluate the performance of relative error estimation, we select three representative estimators from different methodological families: Causal Forest (tree-based) (Athey & Wager, 2019), X-Learner (meta-learner) (Künzel et al., 2019), and TARNet (representation learning) (Shalit et al., 2017a). We estimate their pairwise relative errors and evaluate the estimation performances.

For HTE estimation, the baselines include Causal Forest (Athey & Wager, 2019), meta-learners (X-Learner, S-Learner) (Künzel et al., 2019), double machine learning (Linear DML, Sparse Linear

Table 2: $\delta$ Inference with Different Nuisance

| Nuisance | IHDP | | Twins | |
|---|---|---|---|---|
| | Coverage | Selection | Coverage | Selection |
| Regression | 0.94 | 0.44 | 0.94 | 0.88 |
| Boosting | 0.95 | 0.48 | 0.94 | 0.86 |
| **Ours** | **0.96** | **0.80** | **0.94** | **0.94** |

Table 3: Running Time under Different Settings

| Sample Size | Time (s) | # Candidate Est. | Time (s) |
|---|---|---|---|
| 30 | 2.527 | 2 | 1.0780 |
| 400 | 2.668 | 3 | 3.1321 |
| 500 | 2.739 | 4 | 6.1955 |
| 600 | 3.063 | 5 | 12.2401 |
| 700 | 3.134 | **TARNet** | **2.0306** |

Table 4: Sensitivity analysis on the hyperparameter $\lambda_2$ (weight of constraint loss) for IHDP and Twins datasets. The best hyperparameter values and results are in bold.

| | IHDP | | | | | | Twins | | | | | |
|---|---|---|---|---|---|---|---|---|---|---|---|---|
| Value | $\sqrt{\epsilon_{\text{PEHE}}^{\text{in}}}$ | $\epsilon_{\text{ATE}}^{\text{in}}$ | $\sqrt{\epsilon_{\text{PEHE}}^{\text{out}}}$ | $\epsilon_{\text{ATE}}^{\text{out}}$ | Coverage | Selection | Value | $\sqrt{\epsilon_{\text{PEHE}}^{\text{in}}}$ | $\epsilon_{\text{ATE}}^{\text{in}}$ | $\sqrt{\epsilon_{\text{PEHE}}^{\text{out}}}$ | $\epsilon_{\text{ATE}}^{\text{out}}$ | Coverage | Selection |
| 0.01 | 0.860 | 0.216 | 0.902 | 0.238 | 0.85 | 0.50 | 0.005 | 0.319 | 0.029 | 0.331 | 0.027 | 0.82 | 0.38 |
| 0.1 | 0.800 | 0.142 | 0.837 | 0.158 | 0.91 | 0.61 | 0.05 | 0.289 | 0.016 | 0.292 | 0.015 | 0.82 | 0.84 |
| 0.5 | 0.714 | 0.099 | 0.747 | 0.118 | 0.95 | 0.78 | 0.25 | 0.297 | 0.018 | 0.297 | 0.020 | 0.86 | 0.42 |
| **1** | **0.638** | **0.090** | **0.670** | **0.105** | **0.96** | **0.80** | **0.5** | **0.284** | **0.009** | **0.286** | **0.009** | **0.94** | **0.94** |
| 5 | 0.715 | 0.099 | 0.748 | 0.116 | 0.94 | 0.77 | 2.5 | 0.285 | 0.011 | 0.287 | 0.012 | 0.94 | 0.92 |
| 10 | 0.795 | 0.157 | 0.830 | 0.172 | 0.90 | 0.60 | 5 | 0.289 | 0.028 | 0.290 | 0.026 | 0.80 | 0.86 |
| 100 | 0.801 | 0.156 | 0.836 | 0.170 | 0.90 | 0.60 | 50 | 0.287 | 0.024 | 0.288 | 0.023 | 0.84 | 0.88 |

DML) (Chernozhukov et al., 2024), TARNet (Shalit et al., 2017a), Dragonnet (Shi et al., 2019a), DR-CFR (Hassanpour & Greiner, 2020), SCIGAN (Bica et al., 2020), DESCN (Zhong et al., 2022) and ESCFR (Wang et al., 2023). See Appendix F.10 for training details of hyperparameter tuning.

## 6.2 EXPERIMENTAL RESULTS

**Quality of Relative Error Estimation.** We first evaluate the performance of relative error estimation, comparing different pairs of HTE estimators. In Figures 1 and 2, we present the coverage of the 90% confidence intervals and the accuracy of selecting the better HTE estimator on the test sets, respectively, where TN stands for TARNet, CF stands for Causal Forest, and X stands for X-Learner, and the red dashed line marks the target level of 90%. From these two figures, our method successfully achieves the target coverage, and provide trustworthy advice on the selection across different pairs of HTE estimators. These results demonstrate the validity of our uncertainty quantification and estimator selection.

**Accuracy of the HTE Estimation.** We then evaluate the performance of HTE estimation learned by our novel network and compare it with competing baselines. We average over 100 realizations of our networks in IHDP and 50 realizations in Twins. The results are presented in Table 1. Our proposed method achieves the best performance across all metrics, with the lowest $\sqrt{\epsilon_{\text{PEHE}}}$ and $\epsilon_{\text{ATE}}$ on both datasets. This demonstrates its ability to accurately estimate HTE. In addition, we report results on the **Jobs** dataset in Appendix F.5 due to limited space. We also examine how varying the number of candidate HTE estimators affects the final performance, and report these results in Appendix F.6.

**Comparison with Gao's Method.** Although Gao's work does not propose a concrete learning method, we follow their choice of nuisance estimators (Linear Regression, Boosting) to compute relative errors for reference. We provide the results on the IHDP and the Twins. When plugging conventional nuisance estimators (linear regression and gradient boosting) into the relative error framework, the resulting procedures do achieve nominal coverage (Table 2). Nevertheless, the corresponding variance is so large that the confidence intervals frequently include zero, making it essentially impossible to tell which candidate estimator is superior. These baselines therefore serve as valid but uninformative references. In contrast, our proposed method not only maintains well-calibrated coverage but also delivers much higher selection accuracy, producing confidence intervals that are substantially tighter and practically useful for identifying the winner.

**Running Time under Different Settings.** As the proposed HTE estimator indicates a high level of complexity, we demonstrate experiment results for assessing the computational complexity and scalability of it. We vary sample size and the number of candidate estimators, and a more detailed description is provided in Appendix F.7. It can be seen from Table 3 that our estimator exhibits a runtime that grows approximately linearly with the sample size, but increases super-linearly as the number of candidate estimators in the system expands. Nevertheless, when the system contains only a small number of estimators, our method remains faster than the baseline TARNet.

Table 5: Ablation study results on the IHDP and Twins datasets.

| Training Loss | IHDP | | | | | | Twins | | | | | |
|---|---|---|---|---|---|---|---|---|---|---|---|---|
| | $\sqrt{\epsilon_{\text{PEHE}}^{\text{in}}}$ | $\epsilon_{\text{ATE}}^{\text{in}}$ | $\sqrt{\epsilon_{\text{PEHE}}^{\text{out}}}$ | $\epsilon_{\text{ATE}}^{\text{out}}$ | Coverage | Selection | $\sqrt{\epsilon_{\text{PEHE}}^{\text{in}}}$ | $\epsilon_{\text{ATE}}^{\text{in}}$ | $\sqrt{\epsilon_{\text{PEHE}}^{\text{out}}}$ | $\epsilon_{\text{ATE}}^{\text{out}}$ | Coverage | Selection |
| $\mathcal{L}_{\text{wls}}$ & $\mathcal{L}_{\text{const}}$ | 0.725 | 0.101 | 0.758 | 0.122 | 0.92 | 0.71 | 0.284 | 0.013 | 0.287 | 0.013 | 0.94 | 0.92 |
| $\mathcal{L}_{\text{wls}}$ & $\mathcal{L}_{\text{ce}}$ | 3.495 | 2.879 | 3.531 | 2.900 | 0.88 | 0.14 | 0.319 | 0.028 | 0.328 | 0.026 | 0.82 | 0.14 |
| **Full (Ours)** | **0.638** | **0.090** | **0.670** | **0.105** | **0.96** | **0.80** | **0.284** | **0.009** | **0.286** | **0.009** | **0.94** | **0.94** |

Table 6: Sensitivity Analysis on Propensity Score

| Metric | $(0.05, 0.1^2)$ | $(0.1, 0.1^2)$ | $(0.2, 0.1^2)$ | $(0.05, 0.3^2)$ | $(0.1, 0.3^2)$ | $(0.2, 0.3^2)$ | no noise |
|---|---|---|---|---|---|---|---|
| **Coverage** | 0.88 | 0.94 | 0.96 | 0.88 | 0.94 | 0.80 | 0.96 |
| **Selection** | 0.74 | 0.78 | 0.82 | 0.76 | 0.78 | 0.74 | 0.84 |

**Sensitivity Towards Propensity Score Misspecification.** To test if our evaluation method is sensitive towards the misspecification of propensity score, we conduct sensitivity analysis. Because the true propensity score is required for this experiment, we use simulated data here, and the data-generating process is provided in Appendix F.9. To evaluate the sensitivity of our method to the propensity score, we fix the $t$-head of our proposed neural network to the input propensity score while keeping all other computations unchanged. The input propensity score is constructed by adding Gaussian noise with varying means and variances to the true propensity score. As shown in Table 6 (we provide $(\mu, \sigma^2)$ of the gaussian noise), injecting noise leads to a degradation in the accuracy and validity of the relative-error confidence intervals, but the decline is not substantial. Overall, our method appears to be reasonably robust to perturbations in the propensity score.

**Sensitive Analysis.** The hyperparameter $\lambda_2$ before the constraint loss $\mathcal{L}_{\text{const}}$, $\lambda_1$ before the cross entropy loss $\mathcal{L}_{\text{ce}}$ and the penalty weight $\rho$ in the constraint loss play important roles in our training. In order to explore under which parameters our method has the best performance, we conduct sensitivity analysis experiments. We present the results of $\lambda_2$ in Table 4. We observe that both the performance of HTE estimation and the relative error estimation remain relatively stable across a range of $\lambda_2$ values from 0.5 to 5, indicating robustness to this hyperparameter. However, when $\lambda_2$ is extremely small (e.g., $\lambda_2 = 0.01$), the performance of the proposed method degrades significantly, indicating the importance of the constraint loss $\mathcal{L}_{\text{const}}$. Also, we perform sensitive analysis for $\lambda_1$ and $\rho$, the associated results are provided in the Appendix F.8.

**Ablation Study.** As shown in Section 4.3, the proposed method involves three loss functions: $\mathcal{L}_{\text{wls}}$, $\mathcal{L}_{\text{ce}}$, and $\mathcal{L}_{\text{const}}$. We conduct an ablation study to assess the impact of $\mathcal{L}_{\text{ce}}$ and $\mathcal{L}_{\text{const}}$ on overall performance. The corresponding results are reported in Table 5. Specifically, removing $\mathcal{L}_{\text{const}}$ results in a notable drop in the accuracy of both outcome and relative error estimation, whereas removing $\mathcal{L}_{\text{ce}}$ only causes a moderate decline. These findings highlight the importance of the proposed novel loss $\mathcal{L}_{\text{const}}$, which not only improves HTE estimation accuracy but also facilitates the construction of narrower and more precise confidence intervals for relative error. Besides, the method ($L_{wls}$&$L_{ce}$) can be seen as a method of (Gao, 2025), where the proposed neural network degenerates to TARNet and serves as a conventional nuisance estimator to be used in Gao's structure. From the result of the coverage rate and selection accuracy of relative errors, we can see that our method significantly outperforms this baseline.

# 7 CONCLUSION

In this work, we tackled the challenge of evaluating HTE estimators with reduced reliance on strong modeling assumptions for nuisance components. Building on the relative error framework, we proposed a new loss together with balance regularizers, and embedded them in a Dragonnet-inspired neural architecture to enable more stable and accurate nuisance learning. A remaining limitation is our use of a simple uniform averaging scheme over all estimator pairs blue in the enhanced HTE estimator, which, while stabilizing performance, may underutilize the heterogeneous strengths of individual estimators. Future work includes developing adaptive weighting strategies for the estimator and incorporating a "worst-case performance" perspective to improve robustness in evaluation, potentially by relaxing Assumption 1 along the lines of Huang et al. (2024). Moreover, although HTE characterizes treatment effect heterogeneity with respect to observed covariates, it differs fundamentally from individual-level treatment effects (Wu et al., 2024a). Complementary investigation of ITE (Wu et al., 2025) or joint distribution of potential outcomes (Wu & Mao, 2025) would be desirable to provide a more comprehensive evaluation alongside HTE.

## ACKNOWLEDGEMENTS

This research was supported the National Natural Science Foundation of China (No. 12301370), the BTBU Digital Business Platform Project by BMEC, the Beijing Key Laboratory of Applied Statistics and Digital Regulation, and Academy for Interdisciplinary Studies at Beijing Technology and Business University.

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

## A  NOTATION SUMMARY

Table 7 below summarizes the notation used in the main text.

Table 7: Notation and their meanings.

| Symbol | Meaning |
| --- | --- |
| $A$ | Binary treatment variable |
| $X$ | Pre-treatment covariates |
| $Y$ | Outcome |
| $\tau(x)$ | Individual treatment effect |
| $e(x)$ | Propensity score |
| $\mu_a(x)$ | Outcome regression function, i.e., $\mu_a(x) = \mathbb{E}[Y \mid X = x, A = a]$ for $a = 0, 1$ |
| $\delta(\hat{\tau}_1, \hat{\tau}_2)$ | Relative error between estimator $\hat{\tau}_1$ and $\hat{\tau}_2$ |
| $\check{\delta}(\hat{\tau}_1, \hat{\tau}_2)$ | Estimated relative error between estimator $\hat{\tau}_1$ and $\hat{\tau}_2$ |
| $\check{e}, \check{\mu}_a, \check{\gamma}, \check{\beta}_a$ | Nuisance estimators for propensity score, conditional outcomes and their coefficients |
| $\bar{e}, \bar{\mu}_a, \bar{\gamma}, \bar{\beta}_a$ | Probability limits of propensity score, conditional outcomes and their coefficients |
| $\Phi(X)$ | The shared representation of $X$, defined in Eq. (1) & (2) |

## B  MERITS OF RELATIVE ERROR

There are several advantages of using relative error over absolute error.

- **(1) Weaker condition**. Condition 2 is strictly weaker than Condition 1. Condition 1 requires that *all nuisance parameter estimators* converge to their true values at a rate faster than $n^{-1/4}$. In contrast, Condition 2 imposes a weaker requirement—only that the product of the bias, $(\tilde{\mu}_a(x) - \mu_a(x))(\tilde{e}(x) - e(x))$, converges at a rate of order $o_{\mathbb{P}}(n^{-1/2})$ as well as the nuisance function estimators being consistent. This allows for cases where $\tilde{e}(x)$ converges at a rate of $o_{\mathbb{P}}(n^{-1/5})$ and $\tilde{\mu}_a(x)$ converges at a rate of $o_{\mathbb{P}}(n^{-1/3})$.

- **(2) Easier to compare multiple estimators.** When comparing two estimators $\hat{\tau}_1(x)$ and $\hat{\tau}_2(x)$ in terms of absolute error, although both $\hat{\phi}(\hat{\tau}_1)$ and $\hat{\phi}(\hat{\tau}_2)$ are asymptotically normal, we cannot directly construct a confidence interval for $\hat{\phi}(\hat{\tau}_1) - \hat{\phi}(\hat{\tau}_2)$ due to their dependency (as they use the same test data and share the same nuisance parameter estimates). In contrast, $\hat{\delta}(\hat{\tau}_1, \hat{\tau}_2)$ does not suffer such a problem.

- **(3) Double robustness**. When we replace $o_{\mathbb{P}}(n^{-1/2})$ in Conditions 1 and 2 with $o_{\mathbb{P}}(1)$, both $\hat{\phi}(\hat{\tau}_1)$ and $\hat{\delta}(\hat{\tau}_1, \hat{\tau}_2)$ are consistent (asymptotically unbiased) under their respective conditions. Thus, from Condition 2, $\hat{\delta}(\hat{\tau}_1, \hat{\tau}_2)$ exhibits the property of double robustness, meaning it is a consistent estimator if either $\tilde{e}(x)$ is consistent or $\tilde{\mu}_a(x)$ for $a = 0, 1$ are consistent. However, $\hat{\phi}(\hat{\tau}_1)$ dose not possess this property.

## C  A MISSPECIFIED EXAMPLE

**Example 1** (A misspecified model). *Consider $X$ as a scalar and assume that the true model is $\mu_a(X) = \mathbb{E}(Y|X, A = a) = X^2 \beta_a^*$, which represents the true data-generating mechanism of $Y$ given $(X, A = a)$. However, if we learn $\mathbb{E}(Y|X, A = a)$ using a linear model, i.e. $\mu_a(X, \beta_a) := X\beta_a, \beta_a \in \mathbb{R}$, we introduce an inductive bias, meaning we can never reach the true value of $\beta^*$, even though the estimator may converge. Specifically, denote $\hat{\beta}_a$ as the least-square estimator of $\beta_a$. By the property of least-square estimator, it converges to $\bar{\beta}_a := \mathbb{E}[XX]^{-1}\mathbb{E}[XY]$, regardless of whether $\mu_a(X, \beta_a)$ is correctly specified or not. Since $\mu_a(X, \beta_a)$ is misspecified, $\bar{\beta}_a \neq \beta_a^*$.*

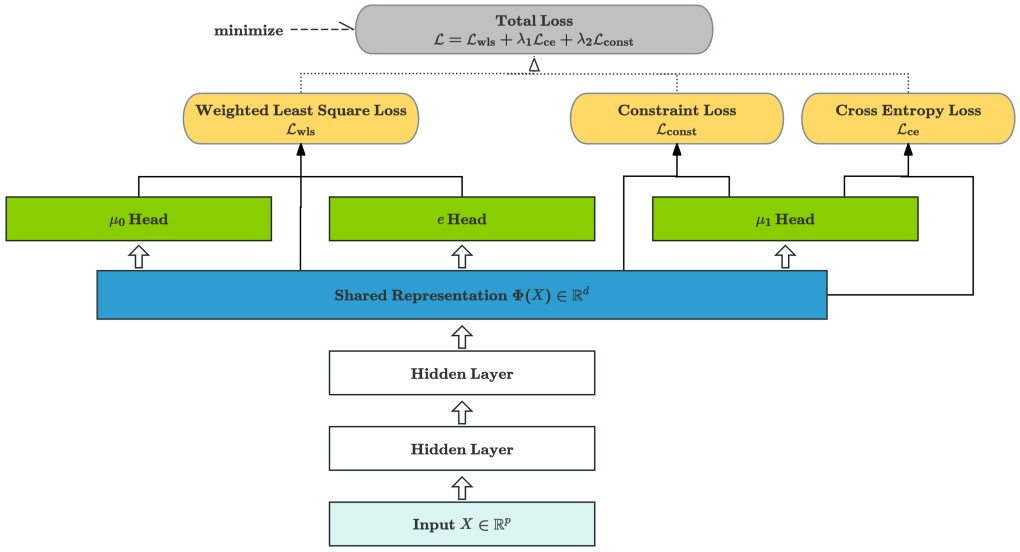

Figure 3: Neural Network Structure

## D    ILLUSTRATION OF NEURAL NETWORK STRUCTURE

Figure 3 shows the schematic structure of our proposed network. The input covariates $X \in \mathbb{R}^p$ are passed through fully connected hidden layers to obtain a shared representation $\Phi(X) \in \mathbb{R}^d$. This representation is fed into three heads: the control outcome head $\mu_0(X)$, the treated outcome head $\mu_1(X)$, and the treatment head $e(X)$. The outcome heads contribute to the weighted least square loss $\mathcal{L}_{\mathrm{wls}}$, the treatment head contributes to the cross entropy loss $\mathcal{L}_{\mathrm{ce}}$, and the shared representation is regularized by the constraint loss $\mathcal{L}_{\mathrm{const}}$. The total objective is given by

$$\mathcal{L} = \mathcal{L}_{\mathrm{wls}} + \lambda_1 \mathcal{L}_{\mathrm{ce}} + \lambda_2 \mathcal{L}_{\mathrm{const}}.$$

## E    PROOF OF THEOREM 1

**Theorem 1.** If the propensity score model is correctly specified, and $\check{\gamma}$, $\check{\beta}_0$ as well as $\check{\beta}_1$ converge to their probability limits at a rate faster than $n^{-1/4}$, then we have

$$\sqrt{n}\{\check{\delta}(\hat{\tau}_1, \hat{\tau}_2; \check{\gamma}, \check{\beta}_0, \check{\beta}_1) - \delta(\hat{\tau}_1, \hat{\tau}_2)\} \xrightarrow{d} \mathcal{N}(0, \sigma^2),$$

where $\sigma^2 = \mathrm{Var}\{\varphi(Z; \bar{u}_0, \bar{u}_1, \bar{e})\}$ and $\xrightarrow{d}$ means convergence in distribution.

*Proof of Theorem 1.* As discussed in Section 4.1, we first show that

$$\check{\delta}(\hat{\tau}_1, \hat{\tau}_2; \check{\gamma}, \check{\beta}_0, \check{\beta}_1) - \check{\delta}(\hat{\tau}_1, \hat{\tau}_2; \bar{\gamma}, \bar{\beta}_0, \bar{\beta}_1) = o_{\mathbb{P}}(n^{-1/2}). \tag{A.1}$$

By a Taylor expansion of $\check{\delta}(\hat{\tau}_1, \hat{\tau}_2; \check{\gamma}, \check{\beta}_0, \check{\beta}_1)$ around $(\bar{\gamma}, \bar{\beta}_0, \bar{\beta}_1)$, we obtain

$$\begin{aligned}
&\check{\delta}(\hat{\tau}_1, \hat{\tau}_2; \check{\gamma}, \check{\beta}_0, \check{\beta}_1) - \check{\delta}(\hat{\tau}_1, \hat{\tau}_2; \bar{\gamma}, \bar{\beta}_0, \bar{\beta}_1) \\
&= \Delta_{\gamma}^{\mathsf{T}}(\check{\gamma} - \bar{\gamma}) + \Delta_{\beta_0}^{\mathsf{T}}(\check{\beta}_0 - \bar{\beta}_0) + \Delta_{\beta_1}^{\mathsf{T}}(\check{\beta}_1 - \bar{\beta}_1) \\
&\quad + O_{\mathbb{P}}((\check{\gamma} - \bar{\gamma})^2 + (\check{\gamma} - \bar{\gamma})(\check{\beta}_1 - \bar{\beta}_1) + (\check{\gamma} - \bar{\gamma})(\check{\beta}_0 - \bar{\beta}_0)),
\end{aligned}$$

where

$$
\Delta_\gamma = -\frac{1}{n} \sum_{i=1}^n 2(\hat{\tau}_1(X_i) - \hat{\tau}_2(X_i))
$$
$$
\times \left( \frac{A_i(1 - \bar{e}(X_i))(Y_i - \bar{\mu}_1(X_i))}{\bar{e}(X_i)} + \frac{(1 - A_i)\bar{e}(X_i)(Y_i - \bar{\mu}_0(X_i))}{1 - \bar{e}(X_i)} \right) \Phi_1(X_i),
$$
$$
\Delta_{\beta_0} = -\frac{1}{n} \sum_{i=1}^n 2(\hat{\tau}_1(X_i) - \hat{\tau}_2(X_i)) \left( 1 - \frac{1 - A_i}{1 - \bar{e}(X_i)} \right) \Phi_2(X_i),
$$
$$
\Delta_{\beta_1} = \frac{1}{n} \sum_{i=1}^n 2(\hat{\tau}_1(X_i) - \hat{\tau}_2(X_i)) \left( 1 - \frac{A_i}{\bar{e}(X_i)} \right) \Phi_2(X_i).
$$

By the condition that $\check{\gamma} = \bar{\gamma} + o_{\mathbb{P}}(n^{-1/4})$, $\check{\beta}_0 = \bar{\beta}_0 + o_{\mathbb{P}}(n^{-1/4})$ and $\check{\beta}_0 = \bar{\beta}_0 + o_{\mathbb{P}}(n^{-1/4})$, we obtain $O_{\mathbb{P}}((\check{\gamma} - \bar{\gamma})^2 + (\check{\gamma} - \bar{\gamma})(\check{\beta}_1 - \bar{\beta}_1) + (\check{\gamma} - \bar{\gamma})(\check{\beta}_0 - \bar{\beta}_0)) = o_{\mathbb{P}}(n^{-1/2})$. Thus, we only need to deal with $\Delta_\gamma^\intercal(\check{\gamma} - \bar{\gamma})$, $\Delta_{\beta_0}^\intercal(\check{\beta}_0 - \bar{\beta}_0)$ and $\Delta_{\beta_1}^\intercal(\check{\beta}_1 - \bar{\beta}_1)$.

Since $\bar{\gamma}$, $\bar{\beta}_0$ and $\bar{\beta}_1$ are the probability limits of $\check{\gamma}$, $\check{\beta}_0$ and $\check{\beta}_1$, respectively, we obtain $\check{\gamma} - \bar{\gamma} = o_{\mathbb{P}}(1)$, $\check{\beta}_0 - \bar{\beta}_0 = o_{\mathbb{P}}(1)$ and $\check{\beta}_1 - \bar{\beta}_1 = o_{\mathbb{P}}(1)$.

Then, it suffices to show that $\Delta_\gamma = O_{\mathbb{P}}(n^{-1/2})$, $\Delta_{\beta_0} = O_{\mathbb{P}}(n^{-1/2})$ and $\Delta_{\beta_1} = O_{\mathbb{P}}(n^{-1/2})$. By CLT, $\Delta_\gamma - \mathbb{E}(\Delta_\gamma) = O_{\mathbb{P}}(n^{-1/2})$, $\Delta_{\beta_0} - \mathbb{E}(\Delta_{\beta_0}) = O_{\mathbb{P}}(n^{-1/2})$ and $\Delta_{\beta_1} - \mathbb{E}(\Delta_{\beta_1}) = O_{\mathbb{P}}(n^{-1/2})$, then, we only need to show that $\mathbb{E}(\Delta_\gamma) = \mathbb{E}(\Delta_{\beta_0}) = \mathbb{E}(\Delta_{\beta_1}) = 0$.

We first deal with $\mathbb{E}(\Delta_\gamma)$.

$$
\mathbb{E}(\Delta_\gamma) = \mathbb{E} \left( -\frac{1}{n} \sum_{i=1}^n 2(\hat{\tau}_1(X_i) - \hat{\tau}_2(X_i)) \left( \frac{A_i(1 - \bar{e}(X_i))(Y_i - \bar{\mu}_1(X_i))}{\bar{e}(X_i)} + \frac{(1 - A_i)\bar{e}(X_i)(Y_i - \bar{\mu}_0(X_i))}{1 - \bar{e}(X_i)} \right) \Phi_1(X_i) \right)
$$
$$
= 2\mathbb{E} \left( (\hat{\tau}_1(X) - \hat{\tau}_2(X)) \left( \frac{A(1 - \bar{e}(X))(Y - \bar{\mu}_1(X))}{\bar{e}(X)} + \frac{(1 - A)\bar{e}(X)(Y - \bar{\mu}_0(X))}{1 - \bar{e}(X)} \right) \Phi_1(X) \right)
$$
$$
= 2\mathbb{E} \left( (\hat{\tau}_1(X) - \hat{\tau}_2(X)) \frac{A(1 - \bar{e}(X))(Y - \bar{\mu}_1(X))}{\bar{e}(X)} \Phi_1(X) \right)
$$
$$
+ 2\mathbb{E} \left( (\hat{\tau}_1(X) - \hat{\tau}_2(X)) \frac{(1 - A)\bar{e}(X)(Y - \bar{\mu}_0(X))}{1 - \bar{e}(X)} \Phi_1(X) \right)
$$
$$
= 0.
$$

The last equation holds by the definition of $\bar{\beta}_0$ and $\bar{\beta}_1$ and the fact that $\Phi_1(X)$ is a sub-vector of $\Phi_2(X)$.

We then deal with $\mathbb{E}(\Delta_{\beta_0})$.

$$
\mathbb{E}(\Delta_{\beta_0}) = \mathbb{E} \left( -\frac{1}{n} \sum_{i=1}^n 2(\hat{\tau}_1(X_i) - \hat{\tau}_2(X_i)) \left( 1 - \frac{1 - A_i}{1 - \bar{e}(X_i)} \right) \Phi_2(X_i) \right)
$$
$$
= -2\mathbb{E} \left( (\hat{\tau}_1(X) - \hat{\tau}_2(X)) \left( 1 - \frac{1 - A}{1 - \bar{e}(X)} \right) \Phi_2(X) \right)
$$
$$
= -2\mathbb{E}_X \left( \mathbb{E} \left( (\hat{\tau}_1(X) - \hat{\tau}_2(X)) \left( 1 - \frac{1 - A}{1 - \bar{e}(X)} \right) \Phi_2(X) \Big| X \right) \right)
$$
$$
= 0.
$$

The last equation holds since the PS model is correct. Finally, we handle $\mathbb{E}(\Delta_{\beta_1})$.

$$\begin{aligned}
\mathbb{E}(\Delta_{\beta_1}) &= \mathbb{E}\left(\frac{1}{n}\sum_{i=1}^{n} 2(\hat{\tau}_1(X_i) - \hat{\tau}_2(X_i))\left(1 - \frac{A_i}{\bar{e}(X_i)}\right)\Phi_2(X_i)\right) \\
&= 2\mathbb{E}\left((\hat{\tau}_1(X) - \hat{\tau}_2(X))\left(1 - \frac{A}{\bar{e}(X)}\right)\Phi_2(X)\right) \\
&= 2\mathbb{E}_X\left(\mathbb{E}\left((\hat{\tau}_1(X) - \hat{\tau}_2(X))\left(1 - \frac{A}{\bar{e}(X)}\right)\Phi_2(X)\Big| X\right)\right) \\
&= 0.
\end{aligned}$$

The last equation holds since the PS model is correct. Therefore, equation A.1 holds.

We then want to show that

$$\sqrt{n}(\check{\delta}(\hat{\tau}_1, \hat{\tau}_2; \bar{\gamma}, \bar{\beta}_0, \bar{\beta}_1) - \delta(\hat{\tau}_1, \hat{\tau}_2)) \to \mathcal{N}(0, \sigma^2). \tag{A.2}$$

By definiton,

$$\begin{aligned}
&\check{\delta}(\hat{\tau}_1, \hat{\tau}_2; \bar{\gamma}, \bar{\beta}_0, \bar{\beta}_1) \\
&= \frac{1}{n}\sum_{i=1}^{n}\{\hat{\tau}_1^2(X_i) - \hat{\tau}_2^2(X_i)\} \\
&\quad - \frac{1}{n}\sum_{i=1}^{n}\left(2\{\hat{\tau}_1(X_i) - \hat{\tau}_2(X_i)\}\cdot\left[\frac{A_i\{Y_i - \bar{\mu}_1(X_i)\}}{\bar{e}(X_i)} + \bar{\mu}_1(X_i) - \frac{(1-A_i)\{Y_i - \bar{\mu}_0(X_i)\}}{1-\bar{e}(X_i)} - \bar{\mu}_0(X_i)\right]\right)
\end{aligned}$$

If model 1 is correct,

$$\begin{aligned}
&\mathbb{E}\left[\{\hat{\tau}_1^2(X_i) - \hat{\tau}_2^2(X_i)\}\right] \\
&- \mathbb{E}\left\{\left(2\{\hat{\tau}_1(X_i) - \hat{\tau}_2(X_i)\}\cdot\left[\frac{A_i\{Y_i - \bar{\mu}_1(X_i)\}}{\bar{e}(X_i)} + \bar{\mu}_1(X_i) - \frac{(1-A_i)\{Y_i - \bar{\mu}_0(X_i)\}}{1-\bar{e}(X_i)} - \bar{\mu}_0(X_i)\right]\right)\right\} \\
&= \mathbb{E}\left[\{\hat{\tau}_1^2(X) - \hat{\tau}_2^2(X)\}\right] \\
&\quad - \mathbb{E}_{X_i}\mathbb{E}\left[\left\{\left(2\{\hat{\tau}_1(X_i) - \hat{\tau}_2(X_i)\}\cdot\left[\frac{A_i\{Y_i - \bar{\mu}_1(X_i)\}}{\bar{e}(X_i)} + \bar{\mu}_1(X_i) - \frac{(1-A_i)\{Y_i - \bar{\mu}_0(X_i)\}}{1-\bar{e}(X_i)} - \bar{\mu}_0(X_i)\right]\right)\right\}\Big| X_i\right] \\
&= \mathbb{E}\left[\{\hat{\tau}_1^2(X) - \hat{\tau}_2^2(X)\}\right] \\
&\quad - \mathbb{E}_{X_i}\mathbb{E}\left[\left\{\left(2\{\hat{\tau}_1(X_i) - \hat{\tau}_2(X_i)\}\cdot\left[\frac{A_i\{Y_i(1) - \bar{\mu}_1(X_i)\}}{\bar{e}(X_i)} + \bar{\mu}_1(X_i)\right.\right.\right.\right. \\
&\qquad\left.\left.\left.\left. - \frac{(1-A_i)\{Y_i(0) - \bar{\mu}_0(X_i)\}}{1-\bar{e}(X_i)} - \bar{\mu}_0(X_i)\right]\right)\right\}\Big| X_i\right] \\
&= \mathbb{E}\left[\{\hat{\tau}_1^2(X) - \hat{\tau}_2^2(X)\}\right] \\
&\quad - \mathbb{E}_{X_i}\mathbb{E}\left[\{(2\{\hat{\tau}_1(X_i) - \hat{\tau}_2(X_i)\}\cdot[\{Y_i(1) - \bar{\mu}_1(X_i)\} + \bar{\mu}_1(X_i) - \{Y_i(0) - \bar{\mu}_0(X_i)\} - \bar{\mu}_0(X_i)])\}| X_i\right] \\
&= \mathbb{E}\left[\{\hat{\tau}_1^2(X) - \hat{\tau}_2^2(X)\}\right] - \mathbb{E}_{X_i}\mathbb{E}[\{(2\{\hat{\tau}_1(X_i) - \hat{\tau}_2(X_i)\}\tau(X_i)| X_i] \\
&= \mathbb{E}\left[\{\hat{\tau}_1^2(X) - \hat{\tau}_2^2(X)\} - 2\{\hat{\tau}_1(X) - \hat{\tau}_2(X)\}\tau(X)\right] \\
&= \delta(\hat{\tau}_1, \hat{\tau}_2).
\end{aligned}$$

If model 2 is correct,

$$
\mathbb{E}\left[\{\hat{\tau}_1^2(X_i) - \hat{\tau}_2^2(X_i)\}\right]
$$

$$
-\mathbb{E}\left\{\left(2\{\hat{\tau}_1(X_i) - \hat{\tau}_2(X_i)\} \cdot \left[\frac{A_i\{Y_i - \bar{\mu}_1(X_i)\}}{\bar{e}(X_i)} + \bar{\mu}_1(X_i) - \frac{(1 - A_i)\{Y_i - \bar{\mu}_0(X_i)\}}{1 - \bar{e}(X_i)} - \bar{\mu}_0(X_i)\right]\right)\right\}
$$

$$
=\mathbb{E}\left[\{\hat{\tau}_1^2(X) - \hat{\tau}_2^2(X)\}\right]
$$

$$
-\mathbb{E}_{X_i}\mathbb{E}\left[\left\{\left(2\{\hat{\tau}_1(X_i) - \hat{\tau}_2(X_i)\} \cdot \left[\frac{A_i\{Y_i - \bar{\mu}_1(X_i)\}}{\bar{e}(X_i)} + \bar{\mu}_1(X_i) - \frac{(1 - A_i)\{Y_i - \bar{\mu}_0(X_i)\}}{1 - \bar{e}(X_i)} - \bar{\mu}_0(X_i)\right]\right)\right\}\middle| X_i\right]
$$

$$
=\mathbb{E}\left[\{\hat{\tau}_1^2(X) - \hat{\tau}_2^2(X)\}\right]
$$

$$
-\mathbb{E}_{X_i}\mathbb{E}\left[\left\{\left(2\{\hat{\tau}_1(X_i) - \hat{\tau}_2(X_i)\} \cdot \left[\frac{A_i\{Y_i(1) - \bar{\mu}_1(X_i)\}}{\bar{e}(X_i)} + \bar{\mu}_1(X_i)\right.\right.\right.\right.
$$

$$
\left.\left.\left.\left.-\frac{(1 - A_i)\{Y_i(0) - \bar{\mu}_0(X_i)\}}{1 - \bar{e}(X_i)} - \bar{\mu}_0(X_i)\right]\right)\right\}\middle| X_i\right]
$$

$$
=\mathbb{E}\left[\{\hat{\tau}_1^2(X) - \hat{\tau}_2^2(X)\}\right]
$$

$$
-\mathbb{E}_{X_i}\mathbb{E}\left[\left\{\left(2\{\hat{\tau}_1(X_i) - \hat{\tau}_2(X_i)\} \cdot \left[\frac{A_i\{\bar{\mu}_1(X_i) - \bar{\mu}_1(X_i)\}}{\bar{e}(X_i)} + \bar{\mu}_1(X_i)\right.\right.\right.\right.
$$

$$
\left.\left.\left.\left.-\frac{(1 - A_i)\{\bar{\mu}_0(X_i) - \bar{\mu}_0(X_i)\}}{1 - \bar{e}(X_i)} - \bar{\mu}_0(X_i)\right]\right)\right\}\middle| X_i\right]
$$

$$
=\mathbb{E}\left[\{\hat{\tau}_1^2(X) - \hat{\tau}_2^2(X)\}\right] - \mathbb{E}_{X_i}\mathbb{E}[\{(2\{\hat{\tau}_1(X_i) - \hat{\tau}_2(X_i)\}\tau(X_i)|X_i]
$$

$$
=\mathbb{E}\left[\{\hat{\tau}_1^2(X) - \hat{\tau}_2^2(X)\} - 2\{\hat{\tau}_1(X) - \hat{\tau}_2(X)\}\tau(X)\right]
$$

$$
=\delta(\hat{\tau}_1, \hat{\tau}_2).
$$

Therefore, when at least one of models 1 or 2 is correct, $\bar{\delta}(\hat{\tau}_1, \hat{\tau}_2; \bar{\gamma}, \bar{\beta}_0, \bar{\beta}_1) - \delta(\hat{\tau}_1, \hat{\tau}_2)$ is the average of i.i.d. observations with mean 0 and variance $\text{Var}\{\varphi(Z; \bar{u}_0, \bar{u}_1, \bar{e})\}$. By CLT, A.2 holds with $\sigma^2 = \text{Var}\{\varphi(Z; \bar{u}_0, \bar{u}_1, \bar{e})\}$.

$\square$

### E.1 PROOF OF PROPOSITION 2

**Proposition 2.** Under the conditions in Theorem 1, a consistent estimator of $\sigma^2$ is

$$
\hat{\sigma}^2 = \frac{1}{n}\sum_{i=1}^{n}\left\{\varphi(Z_i; \check{u}_0, \check{u}_1, \check{e}) - \check{\delta}(\hat{\tau}_1, \hat{\tau}_2; \check{\gamma}, \check{\beta}_0, \check{\beta}_1)\right\}^2,
$$

an asymptotic $(1 - \eta)$ confidence interval for $\delta(\hat{\tau}_1, \hat{\tau}_2)$ is $\check{\delta}(\hat{\tau}_1, \hat{\tau}_2; \check{\gamma}, \check{\beta}_0, \check{\beta}_1) \pm z_{\eta/2}\sqrt{\hat{\sigma}^2/n}$, where $z_{\eta/2}$ is the $(1 - \eta/2)$ quantile of the standard normal distribution.

*Proof of Proposition 2.*

$$
\hat{\sigma}^2 - \sigma^2 = \frac{1}{n}\sum_{i=1}^{n}\left\{\varphi(Z_i; \check{u}_0, \check{u}_1, \check{e}) - \check{\delta}(\hat{\tau}_1, \hat{\tau}_2; \check{\gamma}, \check{\beta}_0, \check{\beta}_1)\right\}^2 - \mathbb{E}\left\{\varphi(Z_i; \bar{u}_0, \bar{u}_1, \bar{e}) - \mathbb{E}\varphi(Z_i; \bar{u}_0, \bar{u}_1, \bar{e})\right\}^2
$$

$$
= \underbrace{\frac{1}{n}\sum_{i=1}^{n}\left\{\varphi(Z_i; \check{u}_0, \check{u}_1, \check{e}) - \check{\delta}(\hat{\tau}_1, \hat{\tau}_2; \check{\gamma}, \check{\beta}_0, \check{\beta}_1)\right\}^2 - \frac{1}{n}\sum_{i=1}^{n}\left\{\varphi(Z_i; \bar{u}_0, \bar{u}_1, \bar{e}) - \mathbb{E}\varphi(Z_i; \bar{u}_0, \bar{u}_1, \bar{e})\right\}^2}_{\textcircled{1}}
$$

$$
+ \underbrace{\frac{1}{n}\sum_{i=1}^{n}\left\{\varphi(Z_i; \bar{u}_0, \bar{u}_1, \bar{e}) - \mathbb{E}\varphi(Z_i; \bar{u}_0, \bar{u}_1, \bar{e})\right\}^2 - \mathbb{E}\left\{\varphi(Z_i; \bar{u}_0, \bar{u}_1, \bar{e}) - \mathbb{E}\varphi(Z_i; \bar{u}_0, \bar{u}_1, \bar{e})\right\}^2}_{\textcircled{2}}.
$$

By the law of large numbers (LLN), ② $\overset{p}{\to} 0$, we only need to deal with ①:

$$\frac{1}{n}\sum_{i=1}^{n}\left\{\varphi(Z_i; \check{u}_0, \check{u}_1, \check{e}) - \check{\delta}(\hat{\tau}_1, \hat{\tau}_2; \check{\gamma}, \check{\beta}_0, \check{\beta}_1)\right\}^2$$

$$=\frac{1}{n}\sum_{i=1}^{n}\left\{\varphi(Z_i; \check{u}_0, \check{u}_1, \check{e}) - \varphi(Z_i; \bar{u}_0, \bar{u}_1, \bar{e}) + \varphi(Z_i; \bar{u}_0, \bar{u}_1, \bar{e}) - \check{\delta}(\hat{\tau}_1, \hat{\tau}_2; \check{\gamma}, \check{\beta}_0, \check{\beta}_1)\right\}^2$$

$$=\underbrace{\frac{1}{n}\sum_{i=1}^{n}\left\{\varphi(Z_i; \check{u}_0, \check{u}_1, \check{e}) - \varphi(Z_i; \bar{u}_0, \bar{u}_1, \bar{e})\right\}^2}_{\text{ⓐ}} + \underbrace{\frac{1}{n}\sum_{i=1}^{n}\left\{\varphi(Z_i; \bar{u}_0, \bar{u}_1, \bar{e}) - \check{\delta}(\hat{\tau}_1, \hat{\tau}_2; \check{\gamma}, \check{\beta}_0, \check{\beta}_1)\right\}^2}_{\text{ⓑ}}$$

$$+\underbrace{\frac{2}{n}\sum_{i=1}^{n}\left\{\varphi(Z_i; \check{u}_0, \check{u}_1, \check{e}) - \varphi(Z_i; \bar{u}_0, \bar{u}_1, \bar{e})\right\}\left\{\varphi(Z_i; \bar{u}_0, \bar{u}_1, \bar{e}) - \check{\delta}(\hat{\tau}_1, \hat{\tau}_2; \check{\gamma}, \check{\beta}_0, \check{\beta}_1)\right\}}_{\text{ⓒ}}.$$

By LLN, ⓐ $\overset{p}{\to} 0$, and ⓒ $\overset{p}{\to} 0$. Similarly, we can obtain

$$\text{ⓑ} = \frac{1}{n}\sum_{i=1}^{n}\left\{\varphi(Z_i; \bar{u}_0, \bar{u}_1, \bar{e}) - \mathbb{E}\varphi(Z_i; \bar{u}_0, \bar{u}_1, \bar{e})\right\}^2 + o_p(1).$$

Therefore ① $\overset{p}{\to} 0$, which leads to $\hat{\sigma}^2 \overset{p}{\to} \sigma^2$.

The asymptotic $1 - \eta$ confidence interval is constructed by the standard theory.

$\square$

# F EXPERIMENTAL DETAILS

## F.1 DATASET DETAILS

**IHDP.** The IHDP dataset is based on a randomized controlled trial conducted as part of the Infant Health and Development Program. The goal is to assess the impact of specialist home visits on children's future cognitive outcomes. Following Hill (and, 2011), a subset of treated units is removed to introduce selection bias, creating a semi-synthetic evaluation setting. The dataset contains 747 samples (139 treated and 608 control), each with 25 pre-treatment covariates. The simulated outcome is the same as that in Shalit et al.(2017) (Shalit et al., 2017a), by setting "A" in the NPCI package (Dorie, 2016).

**Twins.** The Twins dataset is constructed from twin births in the U.S.. For each twin pair, the heavier twin is assigned as the treated unit ($t_i = 1$), and the lighter twin as the control ($t_i = 0$). We extract 28 covariates related to parental, pregnancy, and birth characteristics from the original data and generate an additional 10 covariates following (Wu et al., 2022). The outcome of interest is the one-year mortality of each child. We restrict the analysis to same-sex twins with birth weights below 2000g and without any missing features, yielding a final dataset with 5,271 samples. The treatment assignment mechanism is defined as: $t_i \mid x_i \sim \text{Bern}\left(\sigma(w^\top X + n)\right)$, where $\sigma(\cdot)$ is the sigmoid function, $w^\top \sim \mathcal{U}((-0.1, 0.1)^{38 \times 1})$, and $n \sim \mathcal{N}(0, 0.1)$.

**Jobs.** The Jobs dataset is a standard benchmark in causal inference, originally introduced by LaLonde (1986) (LaLonde, 1986). It evaluates the impact of job training on employment outcomes by combining data from a randomized study (National Supported Work program) with observational records (PSID), following the setup of Smith and Todd (2005) (A. Smith & E. Todd, 2005). The dataset includes 297 treated units, 425 control units from the experimental sample, and 2490 control units from the observational sample. Each record consists of 8 covariates, such as age, education, ethnicity, and pre-treatment earnings. The task is framed as a binary classification problem predicting unemployment status post-treatment, using features defined by Dehejia and Wahba (2002) (Dehejia & Wahba, 2002).

Table 8: Ablation Study on Removing the Shared Representation (IHDP)

| Method | $\sqrt{\epsilon_{\text{PEHE}}^{\text{in}}}$ | $\epsilon_{\text{ATE}}^{\text{in}}$ | $\sqrt{\epsilon_{\text{PEHE}}^{\text{out}}}$ | $\epsilon_{\text{ATE}}^{\text{out}}$ | Coverage | Selection |
|---|---|---|---|---|---|---|
| separate propensity | $1.659 \pm 0.224$ | $0.216 \pm 0.167$ | $1.576 \pm 0.196$ | $0.187 \pm 0.145$ | 0.88 | 0.25 |
| all separate | $7.177 \pm 0.119$ | $7.074 \pm 0.119$ | $7.133 \pm 0.063$ | $7.057 \pm 0.067$ | 0.83 | 0.21 |
| **Ours** | $\mathbf{0.638 \pm 0.138}$ | $\mathbf{0.090 \pm 0.087}$ | $\mathbf{0.670 \pm 0.150}$ | $\mathbf{0.105 \pm 0.099}$ | **0.96** | **0.80** |

Table 9: Ablation Study on Representation Depth and Width (IHDP)

| Setting | $\sqrt{\epsilon_{\text{PEHE}}^{\text{in}}}$ | $\epsilon_{\text{ATE}}^{\text{in}}$ | $\sqrt{\epsilon_{\text{PEHE}}^{\text{out}}}$ | $\epsilon_{\text{ATE}}^{\text{out}}$ | Coverage | Selection |
|---|---|---|---|---|---|---|
| 4 layers | $0.784 \pm 0.249$ | $0.126 \pm 0.099$ | $0.820 \pm 0.279$ | $0.146 \pm 0.121$ | 0.90 | 0.75 |
| 2 layers | $0.705 \pm 0.083$ | $0.116 \pm 0.095$ | $0.739 \pm 0.103$ | $0.136 \pm 0.113$ | 0.94 | 0.73 |
| $2\times$ width | $0.695 \pm 0.156$ | $0.093 \pm 0.074$ | $0.734 \pm 0.188$ | $0.113 \pm 0.099$ | 0.92 | 0.81 |
| $\frac{1}{2}$ width | $0.897 \pm 0.139$ | $0.112 \pm 0.088$ | $0.932 \pm 0.174$ | $0.141 \pm 0.116$ | 0.91 | 0.72 |
| **Ours (original model)** | $\mathbf{0.638 \pm 0.138}$ | $\mathbf{0.090 \pm 0.087}$ | $\mathbf{0.670 \pm 0.150}$ | $\mathbf{0.105 \pm 0.099}$ | **0.96** | **0.80** |

## F.2 JUSTIFICATION FOR SHARED REPRESENTATION

To further examine the role of the shared representation, we conduct an ablation experiment in which the shared feature extractor is removed and the three task-specific heads (two outcome heads and one propensity head) are trained independently. We evaluate two aspects on the IHDP dataset:

1. the estimation accuracy of the nuisance components (measured by in-sample and out-of-sample PEHE and ATE errors), and

2. the validity of the resulting relative-error inference (coverage and selection accuracy).

Table 8 shows that removing the shared representation leads to a substantial degradation in both nuisance estimation and inferential performance. In contrast, our full model consistently achieves the lowest estimation errors and the highest inferential validity. These results highlight that the shared representation is crucial for stable nuisance learning and is an essential component of our method.

## F.3 SENSITIVITY ANALYSIS ON SHARED REPRESENTATION

Although $\Phi(X)$ is data-driven, once the architecture is fixed, a neural network still constitutes a parametric model, and therefore it is important to demonstrate the robustness of our method when the network specification changes substantially.

To assess this, we conduct an additional experiment on the IHDP dataset under the same setting as Table 1. Specifically, we alter the representation network by (i) changing its depth from 3 layers to 2 and 4 layers, and (ii) doubling or halving the hidden dimension of each layer. These modifications create clear and significant deviations from the original architecture.

As shown in Table 9, despite these substantial changes to the model specification, the performance for both nuisance-estimation accuracy and relative-error inference accuracy is remarkably stable. In particular, the validity and efficiency of the relative-error inference exhibit no meaningful deterioration. This provides strong empirical evidence that our method is robust even when $\Phi(X)$ is misspecified.

## F.4 CONVERGENCE ANALYSIS OF THE LOSSES

We report the procedural results under the same settings as Table 1 in the main text. Here, constraint penalty (CP) refers to the first two terms of the penalty in $\mathcal{L}_{\text{const}}$, which penalize violations of the inequality constraints shown in lines 280–284. Negative penalty (NP) corresponds to the last two terms in $\mathcal{L}_{\text{const}}$, which penalize violations of the inequalities in lines 285–286. From Table 10, it is evident that the original conditions are well satisfied: the vast majority of constraints hold strictly, and the remaining few exhibit only negligible numerical discrepancies.

Table 10: Constraint Penalty Statistics Across Datasets

| Dataset | Mean CP | Mean NP | % Zero CP | % CP $< 10^{-5}$ |
|---------|---------|---------|-----------|------------------|
| IHDP | $1.511 \times 10^{-5}$ | 0 | 72% | 95% |
| Twins | $2.604 \times 10^{-8}$ | 0 | 84% | 100% |

Table 11: Performance on the Jobs dataset (in-sample and out-of-sample).

| **Method** | $\mathcal{R}_{\text{pol}}^{\text{in}}$ | $\epsilon_{\text{ATT}}^{\text{in}}$ | $\mathcal{R}_{\text{pol}}^{\text{out}}$ | $\epsilon_{\text{ATT}}^{\text{out}}$ |
|------------|-----------|-----------|-----------|-----------|
| LinDML | $0.158 \pm 0.015$ | $0.019 \pm 0.015$ | $0.183 \pm 0.040$ | $\mathbf{0.053 \pm 0.051}$ |
| SpaDML | $0.150 \pm 0.024$ | $0.131 \pm 0.118$ | $0.165 \pm 0.046$ | $0.144 \pm 0.134$ |
| CForest | $0.114 \pm 0.016$ | $0.025 \pm 0.018$ | $0.155 \pm 0.028$ | $0.058 \pm 0.047$ |
| X-Learner | $0.169 \pm 0.037$ | $0.026 \pm 0.015$ | $0.173 \pm 0.034$ | $\mathbf{0.053 \pm 0.050}$ |
| S-Learner | $0.148 \pm 0.026$ | $0.095 \pm 0.040$ | $0.160 \pm 0.027$ | $0.115 \pm 0.070$ |
| TarNet | $0.141 \pm 0.005$ | $0.183 \pm 0.047$ | $0.145 \pm 0.009$ | $0.190 \pm 0.074$ |
| Dragonnet | $0.230 \pm 0.011$ | $0.021 \pm 0.018$ | $0.143 \pm 0.009$ | $0.172 \pm 0.039$ |
| DRCFR | $0.142 \pm 0.005$ | $0.122 \pm 0.017$ | $0.218 \pm 0.021$ | $0.048 \pm 0.032$ |
| SCIGAN | $0.144 \pm 0.005$ | $0.112 \pm 0.025$ | $0.220 \pm 0.026$ | $0.049 \pm 0.034$ |
| DESCN | $0.192 \pm 0.029$ | $0.098 \pm 0.029$ | $0.143 \pm 0.011$ | $0.065 \pm 0.046$ |
| ESCFR | $0.202 \pm 0.023$ | $0.086 \pm 0.028$ | $0.145 \pm 0.011$ | $0.076 \pm 0.045$ |
| **Ours** | $\mathbf{0.112 \pm 0.019}$ | $\mathbf{0.018 \pm 0.012}$ | $\mathbf{0.131 \pm 0.030}$ | $\mathbf{0.053 \pm 0.039}$ |

## F.5 RESULTS ON JOBS

**Evaluation Metrics.** For the **Jobs** datasets, as there are no counterfactual outcomes, we report the true Average Treatment Effect on the Treated (ATT) and the Policy Risk ($\mathcal{R}_{\text{pol}}$) recommended by Shalit et al.(Shalit et al., 2017b). Specifically, the policy risk can be estimated using only the randomized subset of the Jobs dataset:

$$\hat{\mathcal{R}}\text{pol} = 1 - \left( \frac{1}{|A_1 \cap T_1 \cap E|} \sum_{x_i \in A_1 \cap T_1 \cap E} y_1^{(i)} \cdot \frac{|A_1 \cap E|}{|E|} + \frac{1}{|A_0 \cap T_0 \cap E|} \sum_{x_i \in A_0 \cap T_0 \cap E} y_0^{(i)} \cdot \frac{|A_0 \cap E|}{|E|} \right)$$

where E denotes units from the experimental group, $A_1 = \{x_i : \hat{y}_1^{(i)} - \hat{y}_0^{(i)} > 0\}$, $A_0 = \{x_i : \hat{y}_1^{(i)} - \hat{y}_0^{(i)} < 0\}$, and $T_1, T_0$ are the treated and control subsets, respectively. Since all treated units $T$ belong to the randomized subset $E$, the true Average Treatment Effect on the Treated (ATT) can be identified and computed as:

$$\text{ATT} = \frac{1}{|T|} \sum_{i \in T} y_i - \frac{1}{|C \cap E|} \sum_{i \in C \cap E} y_i$$

where C denotes the control group. We evaluate estimation accuracy using the ATT error: $\epsilon_{\text{ATT}} = \left| \text{ATT} - \frac{1}{|T|} \sum_{i \in T} (f(x_i, 1) - f(x_i, 0)) \right|$.

**Accuracy of the HTE Estimation.** We evaluate the performance of HTE estimation by our network and compare it with baselines mentioned in Section 6. We average over 20 realizations of our network, and the results are presented in Table 11. One can clearly see that our proposed method achieves the best performance across all metrics, having the lowest $\hat{\mathcal{R}}\text{pol}$ and $\epsilon_{\text{ATT}}$ in both training sets and test sets. **Sensitive Analysis and Ablation Study.** We explore which value of $\lambda_1$, $\lambda_2$ and $\rho$ can achieve the best performance. The results are demonstrated in Table 12. We can see that our model is not sensitive to the change of hyperparameters. That is, the performance of HTE estimation remains relatively stable across a range of hyperparameters. For the ablation study presented in Table 13, as that in IHDP and Twins, taking $\mathcal{L}_{\text{ce}}$ off only causes a moderate decline, while removing $\mathcal{L}_{\text{const}}$ brings a whole disaster.

## F.6 THE EFFECT OF THE NUMBER OF CANDIDATE HTE ESTIMATORS

We use the same setting as Table 1 on IHDP. The candidate estimator list is: TARNet, Causal Forest, X-learner, S-learner. It can be seen from Table 14 that increasing the number of candidate estimators

Table 12: Sensitivity analysis on the Jobs dataset with respect to $\lambda_1$, $\lambda_2$, and $\rho$.

| | $\lambda_1$ | | | | | $\lambda_2$ | | | | | $\rho$ | | | |
|---|---|---|---|---|---|---|---|---|---|---|---|---|---|---|
| Value | $\mathcal{R}_{\text{pol}}^{\text{in}}$ | $\epsilon_{\text{ATT}}^{\text{in}}$ | $\mathcal{R}_{\text{pol}}^{\text{out}}$ | $\epsilon_{\text{ATT}}^{\text{out}}$ | Value | $\mathcal{R}_{\text{pol}}^{\text{in}}$ | $\epsilon_{\text{ATT}}^{\text{in}}$ | $\mathcal{R}_{\text{pol}}^{\text{out}}$ | $\epsilon_{\text{ATT}}^{\text{out}}$ | Value | $\mathcal{R}_{\text{pol}}^{\text{in}}$ | $\epsilon_{\text{ATT}}^{\text{in}}$ | $\mathcal{R}_{\text{pol}}^{\text{out}}$ | $\epsilon_{\text{ATT}}^{\text{out}}$ |
| 0.1 | 0.113 | 0.018 | 0.132 | 0.054 | 0.1 | 0.109 | 0.021 | 0.129 | 0.056 | 10 | 0.124 | 0.020 | 0.141 | 0.051 |
| 0.5 | 0.113 | 0.022 | 0.130 | 0.058 | 0.5 | 0.109 | 0.020 | 0.128 | 0.054 | 50 | 0.112 | 0.019 | 0.131 | 0.052 |
| **1** | **0.112** | **0.018** | **0.131** | **0.053** | **1** | **0.112** | **0.018** | **0.131** | **0.053** | **100** | **0.112** | **0.018** | **0.131** | **0.053** |
| 2 | 0.115 | 0.020 | 0.135 | 0.054 | 2 | 0.117 | 0.019 | 0.135 | 0.050 | 200 | 0.115 | 0.020 | 0.132 | 0.053 |
| 10 | 0.121 | 0.027 | 0.140 | 0.060 | 10 | 0.123 | 0.027 | 0.144 | 0.060 | 1000 | 0.114 | 0.020 | 0.133 | 0.052 |

Table 13: ablation studies Jobs

| Training Loss | $\mathcal{R}_{\text{pol}}^{\text{within-s.}}$ | $\epsilon_{\text{ATT}}^{\text{within-s.}}$ | $\mathcal{R}_{\text{pol}}^{\text{out-of-s.}}$ | $\epsilon_{\text{ATT}}^{\text{out-of-s.}}$ |
|---|---|---|---|---|
| $\mathcal{L}_{\text{wls}}$ & $\mathcal{L}_{\text{const}}$ | 0.114 | 0.023 | 0.134 | 0.053 |
| $\mathcal{L}_{\text{wls}}$ & $\mathcal{L}_{\text{ce}}$ | 0.121 | 0.029 | 0.141 | 0.055 |
| **Full (Ours)** | **0.112** | **0.018** | **0.131** | **0.053** |

does not necessarily improve estimation accuracy, possibly because some of the pairs perform poorly and dilute the overall effect.

Table 14: Effect of Varying the Number of Candidate Estimators

| # Estimators | $\sqrt{\epsilon_{\text{PEHE}}^{\text{in}}}$ | $\epsilon_{\text{ATE}}^{\text{in}}$ | $\sqrt{\epsilon_{\text{PEHE}}^{\text{out}}}$ | $\epsilon_{\text{ATE}}^{\text{out}}$ |
|---|---|---|---|---|
| 2 | 0.708 | 0.107 | 0.740 | 0.112 |
| 3 | 0.638 | 0.090 | 0.670 | 0.105 |
| 4 | 0.6703 | 0.1107 | 0.6980 | 0.1227 |
| TARNet (baseline) | 0.896 | 0.279 | 0.920 | 0.266 |

## F.7 DETAILS OF COMPLEXITY ANALYSIS

We conduct two experiments regarding sample size and number of candidate estimators, respectively.

- We varied the sample size by randomly subsampling the IHDP dataset, and measured the running time of our method under different data scales. This demonstrates how the method scales with input size.

- We varied the number of baseline HTE estimators $\hat{\tau}_k$, used to construct the aggregated CATE estimator $\tilde{\tau}(x)$ as described in Section 4.5. Specifically, for each pair $(k, k')$, we used the shared neural network architecture to obtain outcome regressions $\hat{\mu}_1(x; \hat{\tau}_k, \hat{\tau}_{k'})$ and $\hat{\mu}_0(x; \hat{\tau}_k, \hat{\tau}_{k'})$, and then formed $\tilde{\tau}(x; \hat{\tau}_k, \hat{\tau}_{k'})$. The final output $\tilde{\tau}(x)$ is an average over all (or a subset of) such pairs. Since this pairwise process increases computational cost quadratically with the number of baseline estimators, we report running time to illustrate the method's scalability. We also include the runtime of TARNet as a baseline for comparison.

## F.8 EXTENDED SENSITIVE ANALYSIS

In this section we present the results of the sensitive analysis of hyperparameter $\lambda_1$ and $\rho$ in the IHDP and Twins dataset. One can see from Table 15 and Table 16 that our model is robust to the change of $\lambda_1$ and $\rho$, remaining good performance in the HTE estimation as well as relative error prediction.

## F.9 SIMULATED DATA-GENERATING PROCESS

The data-generating process is described as follows: We sample latent covariates $L \sim \mathcal{N}(0, I_m)$ and split them into $I, C, A, D$. Nonlinear structure is introduced via $I \leftarrow \sin(I) - C$, $A \leftarrow (A_1, A_2)$. The treatment assignment depends on $IC = (I, C)$ through $\pi(Z) = \sigma(IC \cdot w_{IC} + \varepsilon)$, $\varepsilon \sim \mathcal{N}(0, 1)$,

Table 15: Sensitivity analysis of $\lambda_1$ on IHDP and Twins datasets.

| | IHDP | | | | | | Twins | | | | | |
|---|---|---|---|---|---|---|---|---|---|---|---|---|
| **Value** | $\sqrt{\epsilon_{\text{PEHE}}^{\text{in}}}$ | $\epsilon_{\text{ATE}}^{\text{in}}$ | $\sqrt{\epsilon_{\text{PEHE}}^{\text{out}}}$ | $\epsilon_{\text{ATE}}^{\text{out}}$ | Coverage | Selection | **Value** | $\sqrt{\epsilon_{\text{PEHE}}^{\text{in}}}$ | $\epsilon_{\text{ATE}}^{\text{in}}$ | $\sqrt{\epsilon_{\text{PEHE}}^{\text{out}}}$ | $\epsilon_{\text{ATE}}^{\text{out}}$ | Coverage | Selection |
| 0.1 | 0.678 | 0.096 | 0.709 | 0.112 | 0.93 | 0.74 | 0.1 | 0.286 | 0.009 | 0.288 | 0.010 | 0.96 | 0.94 |
| 0.25 | 0.693 | 0.096 | 0.724 | 0.113 | 0.93 | 0.75 | 0.25 | 0.285 | 0.010 | 0.287 | 0.010 | 0.94 | 0.94 |
| **0.5** | **0.638** | **0.090** | **0.670** | **0.105** | **0.96** | **0.80** | **0.5** | **0.284** | **0.009** | **0.286** | **0.009** | **0.94** | **0.94** |
| 1 | 0.712 | 0.103 | 0.746 | 0.115 | 0.96 | 0.79 | 1 | 0.285 | 0.013 | 0.287 | 0.014 | 0.94 | 0.92 |
| 2.5 | 1.011 | 0.245 | 1.036 | 0.262 | 0.94 | 0.77 | 2.5 | 0.283 | 0.015 | 0.284 | 0.016 | 0.92 | 0.88 |

Table 16: Sensitivity analysis of $\rho$ on IHDP and Twins datasets.

| | IHDP | | | | | | Twins | | | | | |
|---|---|---|---|---|---|---|---|---|---|---|---|---|
| **Value** | $\sqrt{\epsilon_{\text{PEHE}}^{\text{in}}}$ | $\epsilon_{\text{ATE}}^{\text{in}}$ | $\sqrt{\epsilon_{\text{PEHE}}^{\text{out}}}$ | $\epsilon_{\text{ATE}}^{\text{out}}$ | Coverage | Selection | **Value** | $\sqrt{\epsilon_{\text{PEHE}}^{\text{in}}}$ | $\epsilon_{\text{ATE}}^{\text{in}}$ | $\sqrt{\epsilon_{\text{PEHE}}^{\text{out}}}$ | $\epsilon_{\text{ATE}}^{\text{out}}$ | Coverage | Selection |
| 10 | 0.698 | 0.108 | 0.735 | 0.123 | 0.96 | 0.78 | 10 | 0.299 | 0.015 | 0.306 | 0.015 | 0.92 | 0.62 |
| 50 | 0.711 | 0.098 | 0.745 | 0.116 | 0.95 | 0.79 | 50 | 0.289 | 0.011 | 0.291 | 0.012 | 0.90 | 0.88 |
| **100** | **0.638** | **0.090** | **0.670** | **0.105** | **0.96** | **0.80** | **100** | **0.284** | **0.009** | **0.286** | **0.009** | **0.94** | **0.94** |
| 200 | 0.737 | 0.103 | 0.772 | 0.123 | 0.94 | 0.76 | 200 | 0.286 | 0.012 | 0.288 | 0.013 | 0.94 | 0.92 |
| 1000 | 0.751 | 0.111 | 0.785 | 0.130 | 0.93 | 0.76 | 1000 | 0.284 | 0.010 | 0.285 | 0.011 | 0.92 | 0.94 |

with clipping to $[0.1, 0.9]$, and $T \sim \text{Bernoulli}(\pi(Z))$. Potential outcomes follow $f_0 = \frac{(C,A)^{\circ 1} w_0}{m_C + m_A} + \eta_0$, $f_1 = \frac{(C,A)^{\circ 2} w_1}{m_C + m_A} + \eta_1$, where $\eta_0, \eta_1 \sim \mathcal{N}(0, 0.5)$, yielding $\mu_0 = f_0$ and $\mu_1 = f_1$.

### F.10 MODEL IMPLEMENTATION

We implement all models using PyTorch and optimize them with the Adam optimizer. The key hyper-parameters include the size of each hidden layer, learning rate, the loss coefficients $\lambda_1$, $\lambda_2$, the penalty coefficient $\rho$, and the number of training epochs. These hyperparameters are manually tuned through empirical trials. The search ranges are as follows: hidden layer size in $\{30, 40, 50, 60, 70\}$, learning rate in $\{5 \times 10^{-4}, 10^{-3}, 2 \times 10^{-3}, 3 \times 10^{-3}\}$, $\lambda_1, \lambda_2$ in $\{0.1, 0.25, 0.5, 1, 2\}$, $\rho$ in $\{10, 50, 100, 200\}$, and number of training epochs in $\{700, 800, 900, 1000, 1100\}$.

## G THE USE OF LARGE LANGUAGE MODELS

We used large language models (LLMs), specifically, ChatGPT 5 to polish the writing of our paper. All ideas, methods, experimental designs, and analyses are our own, and the use of LLMs was limited to language editing.

