# OpenReview forum: "A Relative Error-Based Evaluation Framework of Heterogeneous Treatment Effect Estimators"
_ICLR.cc/2026/Conference — ICLR 2026 Poster_

### Official Review · Reviewer_GfzX · 2025-10-18

**Soundness:** 4
**Presentation:** 3
**Contribution:** 3
**Rating:** 8
**Confidence:** 5

**Summary:**

This paper introduces a new *evaluation framework* for HTE estimators, extending a recently proposed relative error–based framework that compares the performance of two estimators. The proposed approach relaxes a key assumption of the prior method, which required all nuisance parameter estimators to be consistent at a rate faster than n-1/4. In contrast, the new framework only requires the propensity score model to meet this condition, allowing the outcome regression and propensity model to be misspecified. Building on this evaluation framework, the authors further develop a new HTE estimation method and evaluate it on some very often used semi-synthetic datasets, including IHDP, Twins and Jobs, demonstrating improved performance compared to several baseline estimators. The paper also provides sensitivity and ablation analyses to support the empirical findings.

**Strengths:**

This is a solid paper that addresses an important and interesting problem. The main strengths are:

- Valuable Problem: The paper focuses on the reliable evaluation of HTE estimators, which is a critical issue for real-world applications. The idea of using "relative error" to compare models is novel and meaningful.

- Strong Theoretical Contribution: The highlight of this paper is its theoretical work. It successfully relaxes the strict assumptions from prior work (Gao, 2025), notably by removing the requirement for the outcome model to be correctly specified. This makes the proposed framework much more robust and practical for real-world scenarios, which is a significant and elegant theoretical improvement.

- Clear and Solid Method: The paper is very clearly written and easy to follow. The proposed new loss function and network architecture are well-motivated and tightly connected to the theoretical derivations. The overall approach feels very solid.

- Thorough Experiments: The authors provide convincing results from experiments on several standard datasets. The experimental design is comprehensive, including ablation and sensitivity analyses, which effectively validate the proposed method's effectiveness.

**Weaknesses:**

- The paper's primary weakness is the unclear practical motivation for using "relative error." While the metric is theoretically interesting for comparing two estimators, its real-world applicability is not well-established. The paper would be significantly stronger if it could provide clear use cases where practitioners would prefer this comparative measure over standard absolute performance metrics like PEHE. This is particularly important for the ICLR audience, which values practical impact.

- The paper is mathematically dense, and while this contributes to its theoretical rigor, it also appears to have led to several typos and minor inconsistencies. To ensure the paper's core contributions are communicated accurately, a thorough proofreading of all definitions, assumptions, and equations is highly recommended.

**Questions:**

- The paper focuses on HTE estimation but only provides formal definitions for ITE and CATE in the problem setup. For clarity, please provide a precise definition of the HTE you are targeting.

- There appears to be a typo in Assumption 1(i). It should likely be the standard unconfoundedness assumption: $(Y_i(0), Y_i(1)) \perp A_i \mid X_i$. Please clarify and correct this.

- Choice of Evaluation Objective: Line 109 states that the goal is to select the estimator with the highest accuracy on a given test dataset. Could the authors elaborate on why this objective was chosen over the more common goal of finding an estimator that accurately models the CATE function over the marginal distribution of covariates, $P(X)$?

- Example 1 illustrates model misspecification, which is a well-understood concept in statistical machine learning. To improve conciseness, this example could be simplified or moved to the appendix.

- The neural network architecture described in Section 4.3 appears to be the Dragonnet architecture. Could the authors clarify the novelty of their proposed network, or explicitly frame it as an adaptation of Dragonnet for their specific loss functions?

- Experimental Comparison: The paper's main theoretical contribution is relaxing the assumptions of Gao (2025). It would be valuable to include a baseline that represents the Gao (2025) approach or can you clarify how the proposed estimator outperform it?

- Minor Typos: Line 173: "violating Assumption 2" should likely refer to "violating Condition 2". Line 216: The text seems to have a typo and should probably read "...the proposed estimator of... ".

---

> ### Author Response · Authors · 2025-11-18
> **Rebuttal by Authors (1/2)**
>
> Thank you very much for your positive evaluation of our paper. Below, we hope that our clarification addresses your concerns.
>
> > **W1:** The paper's primary weakness is the unclear practical motivation for using "relative error." While the metric is theoretically interesting for comparing two estimators, its real-world applicability is not well-established. The paper would be significantly stronger if it could provide clear use cases where practitioners would prefer this comparative measure over standard absolute performance metrics like PEHE. This is particularly important for the ICLR audience, which values practical impact.
>
> **Response to W1:** Thank you for your insightful comments. This point is well taken.
> * In Section 3, we mainly present the key theoretical advantages of using relative error rather than absolute error as the evaluation metric.
>
> * In addition, in Appendix B, we summarize further advantages of using relative error over absolute error, including weaker condition, easier to compare multiple estimators by constructing confidence intervals, and double robustness.
>
> These strengths suggest that using relative error can yield more robust and trustworthy evaluations of HTEs.
>
>
> > **W2:** The paper is mathematically dense, and while this contributes to its theoretical rigor, it also appears to have led to several typos and minor inconsistencies. To ensure the paper's core contributions are communicated accurately, a thorough proofreading of all definitions, assumptions, and equations is highly recommended.
>
> **Response to W2:** Thank you for your helpful suggestions. We have thoroughly proofread the article to improve its readability and clarity, please see our revised version.  Thanks again.
>
> > **Q1:** The paper focuses on HTE estimation but only provides formal definitions for ITE and CATE in the problem setup. For clarity, please provide a precise definition of the HTE you are targeting.
>
> **Response to Q1:** Thank you for pointing this out and we apologize for the lack of clarity. In this paper, HTE is equivalent to CATE, and we will revise it to ensure consistency. Please see our revised version.
>
>
> > **Q2:** There appears to be a typo in Assumption 1(i). It should likely be the standard unconfoundedness assumption: $(Y_i(0), Y_i(1)) \perp  A_i \mid X_i$. Please clarify and correct this.
>
> **Response to Q1:** Thank you for pointing this out; we have revised it accordingly. Please see our revised version.
>
>
> > **Q3:** Choice of Evaluation Objective: Line 109 states that the goal is to select the estimator with the highest accuracy on a given test dataset. Could the authors elaborate on why this objective was chosen over the more common goal of finding an estimator that accurately models the CATE function over the marginal distribution of covariates, $P(X)$?
>
> **Response to Q3:** Thank you for your comments, and we apologize for the lack of clarity. In this article, our goal is exactly to find an estimator aimed at accurately modeling the CATE function. For clarity, we have revised it accordingly. Please see our revised version.
>
> > **Q4:** Example 1 illustrates model misspecification, which is a well-understood concept in statistical machine learning. To improve conciseness, this example could be simplified or moved to the appendix.
>
> **Response to Q4:** Thank you for your helpful suggestions. Since model misspecification is a core concept in our paper and is more of a statistical term than a machine-learning one. Given that this paper is submitted to an ML conference, we are concerned that the audience may not be fully familiar with the term. Therefore, we included additional explanation to improve the clarity of the paper. We agree with your suggestion, and in the revised version, we have simplified the description, deferring the detailed explanation to the Appendix. Please see our revised version.
>
> > **Q5:** The neural network architecture described in Section 4.3 appears to be the Dragonnet architecture. Could the authors clarify the novelty of their proposed network, or explicitly frame it as an adaptation of Dragonnet for their specific loss functions?
>
> **Response to Q5:** Thank you for your comments. We would like to clarify that the novelty of our proposed network lies in the specially designed loss functions. In particular, the differences in the head-specific loss components can be summarized as follows. The explicit expressions for these loss terms can be found in Section 4.2.
>
> | Loss type            | Our network | DragonNet |
> |--------------------------|------------------|----------------|
> | Outcome loss         | Weighted least square loss: $\mathcal{L}_{wls}$  |  Least square loss |
> | Propensity score loss | Cross entropy loss + Constraint loss:  $L_{\mathrm{ce}} + L_{\mathrm{const}}$  | Cross entropy loss only |
>
> The proposed loss fully exploits the connection between the propensity score and outcome regression models, and has more desirable properties.

---

> ### Author Response · Authors · 2025-11-18
> **Rebuttal by Authors (2/2)**
>
> > **Q6:** Experimental Comparison: The paper's main theoretical contribution is relaxing the assumptions of Gao (2025). It would be valuable to include a baseline that represents the Gao (2025) approach or can you clarify how the proposed estimator outperform it?
>
> **Response to Q6:** Thank you for your insightful comments. This point is well taken. We compared Gao (2025)’s method in the following two ways.
>
> - First, in our Ablation study  (Table 3), the method ($L_{wls} \\&  L_{ce}$) can be seen as a method of Gao (2025), where the proposed neural network degenerates to TARNet and serves as a conventional nuisance estimator to  be used in Gao's structure.  From the result of the coverage rate and selection accuracy of relative errors, we can see that our method significantly outperforms this baseline. In the revised version, we highlighted this point for improving clarity.
>
> - Second, in lines 417--419, we state that "Although Gao's work does not propose a concrete learning method, we follow their choice of nuisance estimators (Linear Regression, Boosting) to compute relative errors for reference". In Appendix E.2, Table 5 further indicates the superirity of our method.
>
> > **Q7:** Minor Typos: Line 173: "violating Assumption 2" should likely refer to "violating Condition 2". Line 216: The text seems to have a typo and should probably read "...the proposed estimator of... ".
>
> **Response to Q7:** Thank you for pointing this out; we have revised it accordingly. Please see our revised version.

---

> > ### Comment · Reviewer_GfzX · 2025-11-23
> >
> > Thanks for your detailed response and for using the external page to improve the organization of the paper. The logic is now smoother, and the overall readability has improved.
> >
> > Q5, I understand that you are using a different loss because you are targeting a different learning objective. I raised this question because you referred to it as “the proposed network,” but I noticed that the architecture is the same as Dragonnet. That said, it’s fine since the paper does not aim to contribute a new network architecture, and the newly added sentence makes this point clear. Q6. it's Table 5, but that’s fine.
> >
> > I remain positive about this paper and am keeping my original overall score, while increasing the Presentation score from 3 to 4.

---

> > > ### Author Response · Authors · 2025-11-25
> > >
> > > Thank you for your detailed feedback and we are delighted that we have cleared your previous concerns! Also, thanks for remaining your original overall score of 8, and increasing the Presentation score from 3 to 4 - we really appreciate that!

---

### Official Review · Reviewer_M3dQ · 2025-10-25

**Soundness:** 2
**Presentation:** 1
**Contribution:** 2
**Rating:** 2
**Confidence:** 4

**Summary:**

The paper proposes a way of making an estimator of the relative error of two given CATE estimators robust against the misspecification of the mean outcome nuisance \mu_a, relaxing the constraints of existing methods. Furthermore, the by-product of their algorithm could be used to craft a stronger CATE estimator.

**Strengths:**

- The paper successfully identifies a problem, which is that the existing estimators require asymptotic rates of nuisances no less than n^-¼.
- Through Taylor expansion, the paper shows the conditions under which higher error rate of \mu does not affect the asymptotic rate of n^-½ of the relative error estimator. This provides interesting theoretical insights.
- The paper designs a novel loss that reformulates the equation-based condition as a minimization problem which allows batch gradient descent.
- The paper shows that the method has practical significance in getting stronger CATE models.

**Weaknesses:**

1. The paper has limited novelty compared with existing DR estimator Gao (2025). The fundamental form of estimator remains the same as Gao. And the paper seems to just reconstruct the nuisances to meet certain constraints.
2. The parametrization of e(X) and \mu_a(X) in (1),(2) seem quite arbitrary and lack a decent justification on why these nuisances should share representation.
3. There is NO solution to the moment-based conditions. To this, soft penalties are introduced for finite-sample optimization. Without the correct specification of propensity (which is never met in real world applications) or sensitivity analysis on the propensity score, I doubt the complete formulation might just go meaningless.
4. Misspecification can also come from the representation $\Phi(X)$.
5. Nuisance overfit could be a problem without sample splitting.
6. Typo: line 1017 should be L_const.
7. Experiment: 1) The paper does not perform sensitivity analysis (empirical) on the propensity score, which makes the practical utility doubtful. 2) The paper does not explain how they compute the \tau in real world datasets Twins and Jobs.3) The paper does not provide convergence analysis of the losses, especially L_const and L_ce. The whole method would only make sense if the paper could show, at least empirically, that under finite samples the two losses successfully approximate the original condition.

**Questions:**

1. Your asymptotic CI relies on the correct specification of propensities. What happens when this assumption is violated?
2. How is the ground truth $\tau(x)$ constructed for Twins and Jobs
3. Why does the method not require sample-splitting, a standard procedure in orthogonal estimators?
4. Theorem 1 is only correct when Eq.(4) is satisfied, right? But in your method, you cannot make Eq.4 satisfied. Is there any bound analysis?
5. Why is propensity score easier to estimate than the outcome? (line 172 - 175)
6. The complexity of the method seems high. What is the average time cost of achieving an enhanced estimator using your method? And how does it compare to the time cost of training a baseline CATE estimator?

---

> ### Author Response · Authors · 2025-11-19
> **Rebuttal by Authors (1/4)**
>
> Thank you very much for your efforts in evaluating our paper According to your comments, we have revised the manuscript carefully and thoroughly, _with the major changes marked in red_. Below, we hope that our clarification addresses your concerns.
>
> >**W1:** The paper has limited novelty compared with existing DR estimator Gao (2025). The fundamental form of estimator remains the same as Gao. And the paper seems to just reconstruct the nuisances to meet certain constraints.
>
> **Response to W1:** Thank you for your comments. We would like to clarify that our work is different from Gao (2025), in terms of _the required assumptions, the theoretical framework, the practical feasibility and empirical performance._  The novelty of this work does not lie in simply reconstructing nuisance estimators, but in **expanding relative error's applicability, providing an operational evaluation framework, achieving higher selection accuracy empirically, and serendipitously, introducing a new estimator for CATE**.
>
> ### **(1) Broader Applicability**
>
>  - Gao (2025)'s DR estimator is **not sufficiently robust in practice.** Its validity (inference with confidence intervals) **requires both the outcome regression functions and the propensity score to be correctly specified.** This requirement can be **overly stringent** in real-world applications because the outcome regression model heavily relies on extrapolation and is therefore likely to yield biased estimates.
>
> - Our method, however, **allows for misspecification in the outcome regression functions**. It goes **beyond double robustness**, and provides reliable results even when there exists a significant distributional difference between the treated and control groups. As pointed out in many literatures [1-3], the latter case frequently happens in the real world.
>
>
> ### **(2) Operational Evaluation Framework**
>
> - We provide a reliable operational evaluation framework: practitioners can directly input two pre-trained HTE estimators and a test dataset, and obtain accurate estimates of their relative errors together with formal theoretical guarantees.
>
> - Gao (2025) introduces the concept of relative error. However, they do not provide a practical procedure for  obtaining a reliable estimator. In practice, selecting appropriate nuisance parameters estimators with both practical utility and theoretical guarantees is highly nontrivial and challenging.
>
> ### **(3) Higher Empirical Selection Accuracy**
>
> - In our ablation study (Table 3), the variant using only $L_{\mathrm{wls}}$ and $L_{\mathrm{ce}}$ can be regarded as an implementation of Gao (2025), wherein our proposed network collapses to a TARNet-style architecture and functions as a conventional nuisance estimator within Gao’s framework. From the resulting coverage rates and selection accuracy of the relative errors, it is clear that our full method substantially outperforms this baseline.
>
>  - As shown in Table 5 (Appendix E.2), plugging traditional nuisance estimators (the ones Gao's chose in their paper) leads to significantly low accurate selection accuracy, where their corresponding variances are so large that the confidence intervals frequently include zero. Therefore, they should be treated as valid yet uninformative inference. In contrast, our method demonstrates acuteness towards selecting the true winner.
>
>
> ### **(4) New Estimator for CATE**
>
>  - In Section 5, we provide an enhanced estimator using our proposed neural network structures.
>
>  - Empirically, we show in Table 1 that the proposed estimator achieves marvellous prediction quality, outperforming all baselines.
>
> We hope that the above clarification addresses your concerns and **sincerely invite you to re-evaluate our work in light of our explanation.**
>
>
> > **W2:** The parametrization of $e(X)$ and $\mu_a(X)$ in (1),(2) seem quite arbitrary and lack a decent justification on why these nuisances should share representation.
>
> **Response to W2:** Thank you for raising that concern. We first would like to clarify the following two points:
>
> - **The parametrization of $e(X)$ and $\mu_a(X)$ is highly flexible.** For example, for propensity score model $e(X)$, $\Phi(X)$ can be viewed as the penultimate layer of a neural network, with its output passed through a sigmoid function.
>
> - They share a common representation, which (i) facilitates theoretical analysis of the connections between models and helps establish more desirable theoretical properties; (ii) they are widely used in the literature (see, e.g., [3]) and usually yield more stable numerical results.
>
> In addition, to demonstrate this, **we conduct an additional ablation experiment by removing the shared representation**—allowing the three heads to evolve independently without any shared feature learning. We evaluate two aspects: (1) the estimation accuracy of the nuisance components, and (2) the quality of the relative-error inference. The experiment is carried out on the IHDP dataset.

---

> ### Author Response · Authors · 2025-11-19
> **Rebuttal by Authors (2/4)**
>
> |  | $\sqrt{\epsilon_{\text{PEHE}}^{\text{in}}}$ | $\epsilon_{\text{ATE}}^{\text{in}}$  | $\sqrt{\epsilon_{\text{PEHE}}^{\text{out}}}$| $\epsilon_{\text{ATE}}^{\text{out}}$ | Coverage | Selection |
> | ---- | --- | --- | --- | --- | --- | --- |
> | Share outcome + separate propensity |  1.576 $\pm$ 0.196  | 0.187 $\pm$ 0.145  | 1.659 $\pm$  0.224 | 0.216 $\pm$ 0.167 | 0.88  |  0.25 |
> | Separate outcomes and propensity  |  7.133 $\pm$ 0.063 | 7.057 $\pm$ 0.067  | 7.177 $\pm$ 0.119 | 7.074 $\pm$ 0.119 | 0.83  | 0.21  |
> | Ours | 0.638 $\pm$ 0.138  |  0.090 $\pm$ 0.087 |  0.670 $\pm$ 0.150 | 0.105 $\pm$ 0.099  |  0.96 | 0.80  |
>
> As shown in Table, removing the shared representation consistently harms performance: both the estimation accuracy of the nuisance components and the confidence/validity of the relative error deteriorate. These results indicate that shared representations are essential for achieving stable and reliable inference in our method.
>
>
> > **W3:** There is NO solution to the moment-based conditions. To this, soft penalties are introduced for finite-sample optimization. Without the correct specification of propensity (which is never met in real world applications) or sensitivity analysis on the propensity score, I doubt the complete formulation might just go meaningless.
>
> **Response to W3:** Thanks a lot for your comments.
>
> **Theoretically and conceptually, the assumption of correct specification on propensity score is mild.**
> - First, stating propensity score is correctly specified is not a strong assumption. As $\Phi(X)$ can be adaptively learned from the data, we are likely to gain the true working model given the high prediction efficiency of neural networks.
> - Besides,  our assumption is significantly weaker than Gao's, who asks both potential outcomes and propensity score to be correctly specified. We can never avoid setting up assumptions when evaluating HTE estimators, as we indeed cannot observe any true treatment effects.
>
> Please see our response to Q5 for a more detailed discussion.
>
> **Empirically**, as suggested by the reviewer, **we conduct sensitivity analysis on propensity scores**.  Because the true propensity score is required for this analysis, we use simulated data here. The data-generating process is documented at the end of the paper.
> - To evaluate the sensitivity of our method to the propensity score, we fix the $t$-head of our proposed neural network to the input propensity score while keeping all other computations unchanged. The input propensity score is constructed by adding Gaussian noise with varying means and variances to the true propensity score, followed by clipping to the interval $(0.1, 0.9)$.
> - As shown in the table, injecting noise leads to a degradation in the accuracy and validity of the relative-error confidence intervals, but the decline is not substantial. **Overall, our method appears to be reasonably robust to perturbations in the propensity score.**
>
> | Noise Setting                | Coverage Rate | Selection Accuracy |
> |-----------------------------|---------------|--------------------|
> | $+\mathcal{N}(0.05, 0.1^2)$ |      0.88         |          0.74          |
> | $+\mathcal{N}(0.1, 0.1^2)$  |        0.94       |       0.78             |
> | $+\mathcal{N}(0.2, 0.1^2)$  |        0.96       |        0.82            |
> | $+\mathcal{N}(0.05, 0.3^2)$ |      0.88         |        0.76            |
> | $+\mathcal{N}(0.1, 0.3^2)$  |       0.94        |           0.78         |
> | $+\mathcal{N}(0.2, 0.3^2)$  |         0.80      |           0.74         |
> | no noise                    |    0.96           |         0.84           |
> Rebuttal by Authors (3/4)
>
> > **W4:** Misspecification can also come from the representation $\Phi(X)$.
>
> **Response to W4:** Thank you for your thoughtful review and valuable questions. To assess your concern, we conduct an additional experiment on the IHDP dataset under the same setting as Table 1. Specifically, we alter the representation network by (i) changing its depth from 3 layers to 2 and 4 layers, and (ii) doubling or halving the hidden dimension of each layer. These modifications create clear and significant deviations from the original architecture.
>
> | Setting | $\sqrt{\epsilon_{\text{PEHE}}^{\text{in}}}$ | $\epsilon_{\text{ATE}}^{\text{in}}$ | $\sqrt{\epsilon_{\text{PEHE}}^{\text{out}}}$ | $\epsilon_{\text{ATE}}^{\text{out}}$ | Coverage | Selection |
> |-|-|-|-|-|-|-|
> | 4 layers | 0.784 ± 0.249| 0.126 ± 0.099 | 0.820 ± 0.279 | 0.146 ± 0.121 | 0.90 | 0.75   |
> | 2 layers | 0.705 ± 0.083  | 0.116 ± 0.095 | 0.739 ± 0.103 | 0.136 ± 0.113 | 0.94  | 0.73      |
> | 2× width | 0.695 ± 0.156 | 0.093 ± 0.074 | 0.734 ± 0.188 | 0.113 ± 0.099 | 0.92 | 0.81      |
> | 1/2 width | 0.897 ± 0.139   | 0.112 ± 0.088 | 0.932 ± 0.174 | 0.141 ± 0.116 | 0.91  | 0.72      |
> | **Ours** (original model)      | **0.638 ± 0.138**      | **0.090 ± 0.087**    | **0.670 ± 0.150**      | **0.105 ± 0.099**    | **0.96** | **0.80**  |

---

> ### Author Response · Authors · 2025-11-19
> **Rebuttal by Authors (3/4)**
>
> As shown in the table, despite these substantial changes to the model specification, the performance (especially the estimation of relative error) is remarkably stable. This provides strong empirical evidence that our method is robust towards $\Phi(X)$.
>
> > **W5:** Nuisance overfit could be a problem without sample splitting.
>
> **Response to W5:** Thank you for raising this concern. We would like to clarify that **our proposed methodology does not require sample splitting.** The key derivation in Section 4.1, as well as the proofs of Theorem 1 and Proposition 2 in Section 4.4, are conducted using the full dataset without sample splitting. This is another strength of our method.
>
> > **W6:** Typo: line 1017 should be $L_{const}.$
>
> **Response to W6:** Thank you for pointing this out; we have revised it accordingly. Please see our revised version.
>
> > **W7(1):** The paper does not perform sensitivity analysis (empirical) on the propensity score.
>
> **Response to W7(1):** Please see our response to W3.
>
> > **W7(2):** The paper does not explain how they compute the $ \tau $ in real world datasets Twins and Jobs.
>
> **Response to W7(2):** Thank you for your comments. As described in Appendix E.1, the Twins dataset naturally contains paired potential outcomes for each individual. We only generate the treatment assignment, and the assignment mechanism is fully detailed in Appendix E.1. Therefore, the $\tau$ is directly observed and does not require any additional estimation.
>
> For the Jobs dataset, we never claim to compute $\tau$ nor do we use any metric that requires access to the ground‐truth individual treatment effect. Instead, following the evaluation protocol in [5], we rely on $\mathcal{R}_{\text{pol}}$ and ATT (see Appendix E.3), both of which can be computed without observing counterfactual outcomes.
>
> > **W7(3):** The paper does not provide convergence analysis of the losses, especially $L_{const}$ and $L_{ce}$. The whole method would only make sense if the paper could show, at least empirically, that under finite samples the two losses successfully approximate the original condition.
>
> **Response to W7(3):**  Thank you so much for pointing that out! While convergence guarantees for $L_{const}$ and $L_{ce}$ are theoretically desirable, providing such results is fundamentally challenging. Obtaining non-asymptotic convergence guarantees for constraint-based or surrogate losses typically requires very strong structural assumptions—assumptions that would undermine the very motivation of our relaxed formulation.
>
> Nevertheless, **we could demonstrate empirically that these surrogate losses reliably approximate the target condition under finite samples.**
>
> We report the procedural results under the same settings as Table 1 in the main text. Here, constraint penalty refers to the first two terms of the penalty in $L_{const}$, which penalize violations of the inequality constraints shown in lines 280–284. Negative penalty corresponds to the last two terms in $L_{const}$, which penalize violations of the inequalities in lines 285–286.
>
> | Dataset | Mean Constraint Penalty | Mean Negative Penalty | % Zero Constraint Penalty | % Constriant Penalty < 1e-5 |
> | - | -| -| - | - |
> | IHDP| $1.511 \times 10^{-5}$ | 0 | 72% | 95%|
> | Twins| $2.604 \times 10^{-8}$ |0| 84% | 100%|
>
> **From the table above, it is evident that the original conditions are well satisfied: the vast majority of constraints hold strictly, and the remaining few exhibit only negligible numerical discrepancies.**
>
> > **Q1:** Your asymptotic CI relies on the correct specification of propensities. What happens when this assumption is violated?
>
> **Response to Q1:** Please see our response to W3.
>
> > **Q2:**  How is the ground truth $\tau(X)$ constructed for Twins and Jobs
>
> **Response to Q2:** Please see our response to W7(2).
>
> > **Q3:** Why does the method not require sample-splitting, a standard procedure in orthogonal estimators?
>
> **Response to Q3:** Thank you for your comments. This question is exactly the same as W5; please refer to our response to W5 to avoid redundancy. Thanks again.
>
> > **Q4:** Theorem 1 is only correct when Eq.(4) is satisfied, right? But in your method, you cannot make Eq.4 satisfied. Is there any bound analysis?
>
> **Response to Q4:** Thank you for your insightful comments. Yes,  the condition ``the propensity score model is correctly specified" in Theorem 1 implies that Eq. (4) holds. However, in our method, the Eq. (4) is easy to satisfy. Specifically, Eq. (4) is a condition in an asymptotic sense, i.e., as $n$ goes to infinity.
>
> In finite samples, it is sufficient that the estimator on the left-hand side of Eq. (4) close to zero. This is guaranteed if the propensity score model is correctly specified. The requirement for correctly specified propensity score is a mild condition, given that the propensity score is trained on the entire data (no extrapolation), please see our response to Q5 below for more details.

---

> ### Author Response · Authors · 2025-11-19
> **Rebuttal by Authors (4/4)**
>
> > **Q5:** Why is propensity score easier to estimate than the outcome? (lines 172 - 175)
>
> **Response to Q5:** Thank you for the comment. **There are two key reasons that propensity score is easier to estimate than the outcome regression functions.**
>
> - **Adjusting model specification iteratively by balancing checking.** The essence of the propensity score in causal inference lies in its balancing property [6–8]. Specifically, for any measurable function of the covariates, the expectation in the treated group—weighted by the inverse of the true propensity scores—equals that in the entire population [9–10]. This provides a means of evaluating estimated propensity scores by assessing covariate balance. As in [11], **the propensity score model can be iteratively updated using the following steps:**
>
>   - **Step 1: Model specification and estimation**: specify the model and estimate the propensity scores.
>
>   - **Step 2: Covariate balance checking.** If the balance is inadequate, adjust the model specification and re-estimate the propensity scores; if the balance is satisfactory, proceed with the estimated propensity scores.
>
> - **Without model extrapolation.** The propensity score is trained on the entire data,  so there is no concern about model extrapolation. Consequently, we could use flexible machine learning methods for estimating it, with reduced risk of model misspecification.  In contrast, the outcome regression model $\mu_a(x)$ is learned from the data with $A = a$ and then applied to the entire data. It heavily relies on model extrapolation, because the distribution of covariates in the subgroup with $A = a$ can differ substantially from that in the subgroup with $A=1−a$. In this case, if flexible machine/deep learning methods are adopted, it will increase the risk of model extrapolation.
>
>
> > **Q6:** The complexity of the method seems high. What is the average time cost of achieving an enhanced estimator using your method? And how does it compare to the time cost of training a baseline CATE estimator?
>
> **Response to Q6:** To illustrate the cost of time, **we provide additional experiments for assessing the computational complexity and scalability of the proposed method.**
>
>
> - **(Table 1)**: We varied the **sample size** by randomly subsampling the IHDP dataset, and measured the running time of our method under different data scales. This demonstrates how the method scales with input size.
>
> - **(Table 2)**: We varied the number of baseline HTE estimators $\hat{\tau}_k$, used to construct the aggregated CATE estimator $\tilde{\tau}(x)$ as described in Section 4.5. Specifically, for each pair $(k, k')$, we used the shared neural network architecture to obtain outcome regressions $\hat{\mu}_1(x; \hat{\tau}k, \hat{\tau}{k'})$ and $\hat{\mu}_0(x; \hat{\tau}k, \hat{\tau}{k'})$, and then formed $\tilde{\tau}(x; \hat{\tau}k, \hat{\tau}{k'})$. The final output $\tilde{\tau}(x)$ is an average over all (or a subset of) such pairs. Since this pairwise process increases computational cost quadratically with the number of baseline estimators, **we report running time to illustrate the method's scalability.** We also include the runtime of TARNet as a baseline for comparison.
>
> To summarize, our estimator exhibits a runtime that grows approximately linearly with the sample size, but increases super-linearly as the number of candidate estimators in the system expands (1 -> 3 -> 6 -> 12). Nevertheless, when the system contains only a small number of estimators, our method remains faster than the baseline TARNet.
>
> |Sample Size|Running Time (s)|
> |-|-|
> |30|2.527|
> |400|2.668|
> |500|2.739|
> |600|3.063|
> |700|3.134|
>
> |Number of Baseline Estimators|Running Time (s)|
> |-|-|
> |2|1.0780|
> |3|3.1321|
> |4|6.1955|
> |5|12.2401|
> |**TARNet**|**2.0306**|
>
> ---
>
> Please let us know if we have resolved your concerns – thank you!
>
> **References**
>
>
> [1] Jeong et al. Robust causal inference under covariate shift via worst-case subpopulation treatment effects. ICML 2020.
>
> [2] Li et al. Distribution-free prediction intervals under covariate shift, with an application to causal inference. JASA 2024.
>
> [3] Shi et al. Adapting Neural Networks for the Estimation of Treatment Effects. NeurIPS 2019.
>
> [4] Johansson et al., Generalization Bounds and Representation Learning for Estimation of Potential Outcomes and Causal Effects. JMLR 2022.
>
> [5] Shalit et al. Estimating individual treatment effect: generalization bounds and algorithms. ICML 2017
>
> [6] Rosenbaum et al. The central role of the propensity score in observational studies for causal effects. Biometrika 1983.
>
> [7] Imbens et al. Causal Inference For Statistics Social and Biomedical Science. 2015.
>
> [8] Hernan et al. Causal Inference: What If. 2020.
>
> [9] Imai et al. Covariate balancing propensity score. JRSSB 2014.
>
> [10] Sant'Anna et al. Covariate distribution balance via propensity scores. JoE 2022.
>
> [11] Belitser et al. Measuring balance and model selection in propensity score methods. 2011.

---

> > ### Comment · Reviewer_M3dQ · 2025-11-24
> >
> > Thank you for your clarification. The additional effort in experiments shows the validity of the proposed constrained optimization. The explanation on why is propensity scores are easier to estimate makes sense and is well-supported by existing researches. However, some confusions are still not resolved:
> >
> > 1. Gao's condition does not require consistency for both models. To argue that super-fast n^-0.5 rate rate is difficult for a single nuisance estimator, references are needed to back the argument.
> > 2. Although the author claims significant improvement in practicality, new conditions are introduced in theorem 1. It seems the paper essentially swap the tolerance to outcome error with a convergence assumptions on the trained parameter $\beta_0, \beta_1,\gamma$. These new conditions, do not seem to be easier than Gao's assumption. The reason is that learning these parameters will depend on the representation $\Phi$. In practice, this is influenced by the capacity of the network architecture, which varies case by case. To find the correctly specified architecture is not practically easy.
> > 3. I still think the paper is not yet ready (e.g. some standard datasets like ACIC are missing).

---

> > > ### Author Response · Authors · 2025-11-25
> > > **Clarification of misunderstandings on Gao's and our arguments**
> > >
> > > Thank you for your careful reading and appreciation on our empirical results showing the validity of the proposed constrained optimization, as well as our explanation on why is propensity scores are easier to estimate.
> > >
> > > Meanwhile, it seems there may exist some misunderstandings on our work. We hope our clarifications and modifications would make the message clearer and we would appreciate your further feedback!
> > >
> > > > 1. Gao's condition does not require consistency for both models. To argue that super-fast $n^{-1/2}$ rate rate is difficult for a single nuisance estimator, references are needed to back the argument.
> > >
> > > **Response to Q1:** Thank you for your comments, but it seems there is a misunderstanding.
> > >
> > > - First, **Gao's condition does require consistency for both models to be $\sqrt{n}$-consistent**. In Gao (2025), condition (b) in Theorem 4.1 (arXiv version) / Theorem 2 (AISTATS version) explicitly states
> > > $\|\tilde{\mu}_1(X) - \mu_1(X)\|_2= o_p(1),\
> > > \|\tilde{\mu}_0(X) - \mu_0(X)\|_2= o_p(1),\
> > > \|\tilde{e}(X) - e(X)\|_2 = o_p(1)$.
> > > - Second, **we do not claim a "super-fast $n^{-1/2}$ rate rate for a single nuisance estimator".** Instead, for the evaluation of HTE, **we require that the estimator of the relative error be $\sqrt{n}$-consistent and asymptotically normal,** so that we can construct valid confidence intervals to screen out superior HTE estimators with high probability. To achieve $\sqrt{n}$-consistentency and asymptotic normality, **Gao (2025)'s method requires that all nuisance parameter estimators (propensity score and outcome regression models) are consistent at a rate faster than  $n^{-1/4}$,** which is difficult to satisfy in real-world applications. This limitation is the core contribution for our paper.
> > >
> > > > 2. Although the author claims significant improvement in practicality, new conditions are introduced in theorem 1. It seems the paper essentially swap the tolerance to outcome error with a convergence assumptions on the trained parameter $\beta_0, \beta_1,\gamma$. These new conditions, do not seem to be easier than Gao's assumption. The reason is that learning these parameters will depend on the representation $\Phi$. In practice, this is influenced by the capacity of the network architecture, which varies case by case. To find the correctly specified architecture is not practically easy.
> > >
> > > **Response to Q2:** Thank you for your comments! This may also involve a misunderstanding. We would like to clarify the two following points.
> > >
> > > **First, the requirement on convergence assumptions on the trained parameters $\check \beta_0, \check \beta_1, \check \gamma$ is very mild and does not require correct model specification or the representation $\Phi$.** Here are the reasons:
> > > - In Theorem 1, we require that "$\check \gamma, \check \beta_0,$ and  $\check \beta_1$ converge to their probability limits at a rate faster than $n^{-1/4}$."
> > >   - The key point is that these limits are their probability limits, not the true values.
> > >   - This distinction is conceptually fundamental. As explained in lines 204–204 and 325–328 of the revised manuscript, $\check{\gamma}$, $\check{\beta}_0$, and $\check{\beta}_1$ always converge to their respective probability limits, regardless of whether the models are correctly specified.
> > >   - For a given network architecture, **a different representation would only change the definition of the probability limit of nuisance parameters, but does not affect the convergence rate.**
> > >
> > > **Second, our conditions are fundamentally different from those in Gao(2025).** Gao(2025)'s method achieves $\sqrt{n}$-consistency only when both the propensity score model and the outcome regression model are correctly specified, and when $\check{\gamma}$, $\check{\beta}_0$, and $\check{\beta}_1$ converge to their true values at a rate faster than $n^{-1/4}$ (see lines 165–167 of the revised manuscript for details).
> > >
> > > In summary,  the proposed method is not merely doubly robust but in fact beyond (or superior to) standard double robustness. This is because our estimator attains $\sqrt{n}$-consistency as long as the propensity score model is correctly specified, even if the outcome regression model is misspecified (thus beyond double robustness). In contrast, previous method achieves $\sqrt{n}$-consistency only when both the propensity score model and the outcome regression model are correctly specified.

---

> ### Author Response · Authors · 2025-11-25
> **More results on ACIC and News datasets**
>
> > 3. I still think the paper is not yet ready (e.g. some standard datasets like ACIC are missing).
>
> **Response to Q3:**
> Thanks for the great suggestion! We kindly remind that we have validate the effectiveness of our method on IHDP, Jobs, and Twins datasets. In light of your suggestion, **we further add experiments on ACIC and News (Johansson et al., ICML 16) under the same setting** as Figure 1, using X-Learner, S-Learner, and Causal Forest as the three candidate estimators. We report both the coverage rate and selection accuracy of the relative error. The results are summarized in the following tables.
>
> ACIC coverage rate:
>
> |                | Min   | 25%    | 50%    | 75%    | Max   |
> |-------|-------|--------|--------|--------|-------|
> | S-Learner v.s. X-Learner    | 0.68  | 0.86   | 0.90   | 0.93   | 0.99  |
> | S-Learner v.s. Causal Forest    | 0.83  | 0.88 | 0.90  | 0.93   | 0.98  |
> | X-Learner v.s. Causal Forest    | 0.77  | 0.88   | 0.92   | 0.94 | 0.99  |
>
> ACIC selection accuracy:
> |  | Min   | 25%    | 50%    | 75%    | Max   |
> |-----------------------------|-------|--------|--------|--------|-------|
> | S-Learner v.s. X-Learner   | 0.58  | 0.68 | 0.75   | 0.80   | 0.91  |
> | S-Learner v.s. Causal Forest  | 0.60  | 0.66   | 0.71  | 0.78   | 0.88  |
> |  X-Learner v.s. Causal Forest   | 0.64 | 0.78   | 0.83  | 0.87 | 0.92  |
>
> News coverage rate:
>
> |                | Min   | 25%    | 50%    | 75%    | Max   |
> |-------|-------|--------|--------|--------|-------|
> | S-Learner v.s. X-Learner    | 0.82  | 0.88   | 0.93   | 0.96   | 1.00  |
> | S-Learner v.s. Causal Forest    | 0.81  | 0.87 | 0.90  | 0.92   | 0.96  |
> | X-Learner v.s. Causal Forest    | 0.82  | 0.86   | 0.9   | 0.93 | 0.99  |
>
> News selection accuracy:
> |  | Min   | 25%    | 50%    | 75%    | Max   |
> |-----------------------------|-------|--------|--------|--------|-------|
> | S-Learner v.s. X-Learner   | 0.53  | 0.64 | 0.70   | 0.74   | 0.83  |
> | S-Learner v.s. Causal Forest  | 0.70  | 0.78   | 0.83  | 0.86   | 0.91  |
> |  X-Learner v.s. Causal Forest   | 0.60 | 0.66   | 0.73  | 0.78 | 0.89  |
>
> As shown, our method also performs strongly on both ACIC and News, supporting the main text of our paper and demonstrating the generality and applicability of our approach across different 5 datasets.
>
> ***
>
> Thank you for your thoughtful comments and remarks and we’d highly appreciate your further feedback!

---

> ### Comment · Reviewer_M3dQ · 2025-11-27
>
> Thank you for the explanation.
>
> **On the theory**
>
> I do not think the term “reliable” is warranted in the introduction. The core theoretical contribution relies on the overidentified system in Eq. (4), which the paper itself acknowledges to be infeasible in finite samples. To address this, the paper proposes a soft-penalized optimization. However, no theoretical properties of this optimization are established (e.g., existence, uniqueness, or convergence). Given that the optimization problem is highly non-convex, the presence of multiple minimizers would prevent meaningful convergence guarantees. In this sense, it is difficult to justify describing the resulting estimator as “reliable.”
>
> A further concern is that, in practice, the propensity score will almost never be correctly specified. This implies that Eq. (4) is essentially infeasible already at the population level. Relaxing an essentially infeasible population equation into a finite-sample optimization problem is conceptually problematic. Yet Theorem 1 assumes that, at the probability limit (if it exists), the moments are approximately satisfied. It also has downstream implications for whether the variance sigma^2 can be regarded as faithful. This reinforces my view that calling the method “reliable” is too strong.
>
> **On the "enhanced" method**
>
> I am also concerned about the “enhanced method.” To my understanding, it explicitly relies on modeling the potential outcomes $\mu_0, \mu_1$ (in a TARNet like architecture). In much of the SOTA literature on heterogeneous treatment effects, an explicit inductive bias is imposed that the HTE function is simpler (or lower dimensional) than the two potential outcome functions themselves (see, for example, work by Alicia Curth). By contrast, the proposed enhanced method appears to require modeling the full potential outcomes, which seems at odds with the SOTA perspective (e.g., DR-learner) and may undermine some of the practical advantages typically sought in HTE estimation methods.
>
> **Presentation and writing**
>
> Finally, there are still many typos and language issues throughout the paper. For example:
> - “split … sample” instead of “sample splitting,”
> - “correctly specification” instead of “correct specification,”
> - “equation Eq.” (redundant), etc.
>
> There is also occasional confusion between HTE estimation and HTE evaluation in the writing. I recommend a careful proof-reading and clearer separation of these two concepts.

---

> > ### Author Response · Authors · 2025-12-01
> > **Rebuttal by Authors (1/2)**
> >
> > Dear Reviewer M3dQ,
> >
> > Thank you for your feedback. We have carefully considered your points and would like to provide the following clarifications.
> >
> > > **Q1: On the theory.**
> >
> > > **Q1(1):** I do not think the term “reliable” is warranted in the introduction. The core theoretical contribution relies on the overidentified system in Eq. (4), which the paper itself acknowledges to be infeasible in finite samples. To address this, the paper proposes a soft-penalized optimization. However, no theoretical properties of this optimization are established (e.g., existence, uniqueness, or convergence). Given that the optimization problem is highly non-convex, the presence of multiple minimizers would prevent meaningful convergence guarantees. In this sense, it is difficult to justify describing the resulting estimator as “reliable.”
> >
> > **Response to Q1(1):**  Thank you for your comments. In response to your comments, we would like to describe our article from the perspective of balancing.
> > - **Essentially, the last two terms in Eq. (4) are balancing moment conditions on propensity scores.** They ensure the balancing properties of the propensity score in the treated group, the control group, and the overall population.
> > - **Using balancing moment conditions in an overidentified system has a long history in the estimation of propensity scores.** For example,
> >   - In Section 2.2 of [1], the authors impose overidentified moment conditions that exceed the number of parameters in the propensity score model;
> >   - In Section 2.2 of [2], the authors impose infinite-dimensional moment conditions for estimating propensity scores.
> >   - In Section 2 of [3], the authors propose balancing the covariates in  a reproducing-kernel Hilbert space for estimating propensity scores.
> >
> > **All of the literature [1–3] demonstrates the ``reliable" performance of propensity scores estimated using balancing moment conditions.**
> >
> >
> > > **Q1(2):**  A further concern is that, in practice, the propensity score will almost never be correctly specified. This implies that Eq. (4) is essentially infeasible already at the population level. Relaxing an essentially infeasible population equation into a finite-sample optimization problem is conceptually problematic. Yet Theorem 1 assumes that, at the probability limit (if it exists), the moments are approximately satisfied. It also has downstream implications for whether the variance sigma^2 can be regarded as faithful. This reinforces my view that calling the method “reliable” is too strong.
> >
> > **Response to Q1(2):**  We would like to further clarify the subtle **distinctions between the true propensity scores in an infinite population and the estimated propensity scores** in finite samples.
> >   - The true propensity scores satisfy the balancing property: for any measurable function of the covariates, the expectation in the treated group—weighted by the inverse of the true propensity scores—equals that in the entire population [1–2]. Thus, true propensity scores naturally satisfy infinite-dimensional moment conditions.
> >   - However, the true propensity scores are unknown and require estimation from finite samples. As you noted, in practice, the propensity score model is almost never correctly specified. Therefore, in finite samples, similar to literature [1–3], we can aim to ensure that the balancing property holds as well as possible when flexible models are used. This is exactly what we do in our article.
> >   - **In our article, the last two terms in Eq. (4) naturally hold for true propensity scores. For the estimated propensity scores, we aim to ensure that they hold as closely as possible.**
> >
> > Finally, regarding the comment that "in practice, the propensity score will almost never be correctly specified. This implies that Eq. (4) is essentially infeasible already at the population level." **We slighly disagree with the last point**, as it somewhat conflates theory and practice. In theory, almost all propensity score-based methods, when establishing large-sample properties at the population level, require certain conditions on model specification (or their analogs). **In practice, all models are wrong, and we cannot ensure that anything holds absolutely. Nevertheless, this does not mean that theoretical analysis is useless (under the propensity model correctly specification). It still provides valuable insights on algorithm design, ways to improve performance, and how the method behaves under proper conditions.**

---

> ### Author Response · Authors · 2025-12-01
> **Rebuttal by Authors (2/2)**
>
> > **Q2: On the "enhanced" method.**  I am also concerned about the “enhanced method.” To my understanding, it explicitly relies on modeling the potential outcomes $\mu_0, \mu_1$ (in a TARNet like architecture). In much of the SOTA literature on heterogeneous treatment effects, an explicit inductive bias is imposed that the HTE function is simpler (or lower dimensional) than the two potential outcome functions themselves (see, for example, work by Alicia Curth). By contrast, the proposed enhanced method appears to require modeling the full potential outcomes, which seems at odds with the SOTA perspective (e.g., DR-learner) and may undermine some of the practical advantages typically sought in HTE estimation methods.
>
> **Response to Q2:** We appreciate the reviewer's insightful comment. We agree that the enhanced estimator relies on a separate estimation of $\mu_0$ and $\mu_1$ in a TARNet-like architecture. This design choice is **intentional**: our primary goal in this paper is *not* to propose a new HTE estimator, but to learn the relative error, whose definition and estimation require the potential outcome predictions rather than only the CATE function.
>
> Importantly, the enhanced estimator is an **ancillary** finding that emerged during experimentation and **is not part of the three main contributions stated in our Introduction**. We include it mainly because, empirically, this structue yields improved performance compared with a standard TARNet and many commonly used baselines. In other words, its purpose is illustrative rather than foundational: the method is not meant to compete with or contradict SOTA approaches such as DR-learner.
>
> **Our contribution is the evaluation framework, and the enhanced estimator is presented only as a by-product demonstrating how the insights from relative-error modeling may guide estimation.**
>
>
> **Q3: Presentation and writing.**
>
> **Response to Q3:** Thank you for pointing these out. We have revised it accordingly, and the changes are marked in $\textcolor{blue}{blue}$. Please see our revised version in OpenReview.
>
>
> **We hope these clarifications have fully addressed your concerns.**
>
> ---
>
> **References**
>
>
> [1] Kosuke Imai and Marc Ratkovic. Covariate balancing propensity score. Journal of the Royal Statistical Society (Series B), 76(1):243–263, 2014.
>
> [2] Pedro H. C. Sant'Anna, Xiaojun Song, and Qi Xu, Covariate distribution balance via propensity scores, Journal of Applied Econometrics, 37(6): 1093-1120, 2022.
>
> [3] Wong, R. K. W., and Chan, K. C. G. (2018), Kernel-based covariate functional balancing for observational studies, Biometrika, 105(1), 199–213.

---

### Official Review · Reviewer_eY6Y · 2025-11-01

**Soundness:** 4
**Presentation:** 4
**Contribution:** 3
**Rating:** 8
**Confidence:** 4

**Summary:**

This paper extends the work of (Gao 2025) on the use of relative error for heterogeneous treatment effect estimation. Its main contribution is that it offers a relaxation of the requirement that the nuisance parameter estimators must be consistent. Instead, it is shown that robust estimation can be achieved if the a correctly specified propensity score estimator is provided. Moreover, the theoretical results lead to the development of an HTE method, while both theoretical and experimental results support the arguments of the paper.

**Strengths:**

- This paper addresses a very important yet understudied problem, and an effective practical solution for the evaluation of HTE estimators is very valuable
 - The paper offers significant theoretical contributions offering theoretical guarantees on estimator consistency even with misspecified regression models
 - In addition to theoretical results on the evaluation of HTE estimators, the paper offers a learning algorithm and method for HTE estimation
 - The empirical evaluation on datasets available in the literature is comprehensive

**Weaknesses:**

- The need for a correctly specified propensity score model still remains a strong assumption, which may not hold or be guaranteed in practice in many cases.
 - One would expect a deeper evaluation on the comparison of the proposed method against the method by Gao 2025, as well as an empirical evaluation of what happens when the propensity score estimator is misspecified.
 - There is some confusion regarding the terminology, since there is mention of "balance regularizers" in the beginning but later this terminology is abandoned, where $\mathcal{L}_{\text{const}}$ is mentioned

**Questions:**

- Is there any theoretical or empirical evidence of what happens when the propensity score estimation model is misspecified?
- How do results depend on the number of candidate estimators?

---

> ### Author Response · Authors · 2025-11-20
> **Rebuttal by Authors (1/2)**
>
> Thank you very much for your positive evaluation of our paper. According to your comments, we have revised the manuscript carefully and thoroughly, with the major changes marked in red. Below, we hope that our clarification addresses your concerns.
>
> > **W1:** The need for a correctly specified propensity score model still remains a strong assumption, which may not hold or be guaranteed in practice in many cases.
>
> **Response to W1:**  Thank you for your insightful comments.
> We would like to clarify that this is not a major concern, either theoretically or empirically.
>
> **Theoretically and conceptually,** the assumption of correct specification on propensity score is mild.
> - **First, correct specification for propensity score is not a strong assumption.** As $\Phi(X)$ can be adaptively learned from the data, we are likely to gain the true working model using flexible neural networks. In addition, we could adjust the model specification iteratively by balancing checking as follows:
>   - Step 1: **Model specification and estimation:** specify the model and estimate the propensity scores.
>   - Step 2: **Covariate balance checking** [1-3]. If the covariate balance is inadequate, adjust the model specification and re-estimate the propensity scores; if the balance is satisfactory, proceed with the estimated propensity scores.
> - Besides,  our assumption is **significantly weaker** than Gao's, who asks both potential outcomes and propensity score to be correctly specified. We can never avoid setting up assumptions when evaluating HTE estimators, as we indeed cannot observe any true treatment effects.
>
> **Empirically, we conduct additional sensitivity analysis on propensity scores.**  Because the true propensity score is required for this analysis, we use simulated data here. The data-generating process is documented at the end of the paper.
> - To evaluate the sensitivity of our method to the propensity score, we fix the $t$-head of our proposed neural network to the input propensity score while keeping all other computations unchanged. The input propensity score is constructed by adding Gaussian noise with varying means and variances to the true propensity score, followed by clipping to the interval $(0.1, 0.9)$.
> - As shown in the table, injecting noise leads to a degradation in the accuracy and validity of the relative-error confidence intervals, but the decline is not substantial. **Overall, our method appears to be reasonably robust to perturbations in the propensity score.**
>
> | Noise Setting                | Coverage Rate | Selection Accuracy |
> |-----------------------------|---------------|--------------------|
> | $+\mathcal{N}(0.05, 0.1^2)$ |      0.88         |          0.74          |
> | $+\mathcal{N}(0.1, 0.1^2)$  |        0.94       |       0.78             |
> | $+\mathcal{N}(0.2, 0.1^2)$  |        0.96       |        0.82            |
> | $+\mathcal{N}(0.05, 0.3^2)$ |      0.88         |        0.76            |
> | $+\mathcal{N}(0.1, 0.3^2)$  |       0.94        |           0.78         |
> | $+\mathcal{N}(0.2, 0.3^2)$  |         0.80      |           0.74         |
> | no noise                    |    0.96           |         0.84           |
>
> > **W2:** One would expect a deeper evaluation on the comparison of the proposed method against the method by Gao 2025, as well as an empirical evaluation of what happens when the propensity score estimator is misspecified.
>
> **Response to W2:** Thank you for your comment. We compared Gao (2025)’s method in the following two ways.
> - First, in our **Ablation study  (Table 3)**, the method $ L_{wls} + L_{ce} $ can be seen as a method of Gao (2025), where the proposed neural network degenerates to TARNet and serves as a conventional nuisance estimator to  be used in Gao's structure.  From the result of the coverage rate and selection accuracy of relative errors, we can see that our method significantly outperforms this baseline. In the revised version, we highlighted this point for improving clarity.
> - Second, in lines 417--419, we state that "Although Gao's work does not propose a concrete learning method, we follow their choice of nuisance estimators (Linear Regression, Boosting) to compute relative errors for reference". In **Appendix E.2, Table 5** further indicates the superiority of our method.
>
> For the case when propensity score is misspecified, please refer to the additional experiments provided in our response to W1.

---

> ### Author Response · Authors · 2025-11-20
> **Rebuttal by Authors (2/2)**
>
> > **W3:** There is some confusion regarding the terminology, since there is mention of "balance regularizers" in the beginning but later this terminology is abandoned, where $\mathcal{L}_{const}$ is mentioned
>
> **Response to W3:** Thank you for the constructive suggestions to help us improve the readability of our paper! We refer the loss $L_{const}$ to as balance regularizer in the introduction to avoide the use of mathematical formulas. Following your suggestion, in the revised version, we added the terminology in Section 4.2 when $L_{const}$ appears. Thanks again.
>
> > **Q1:** Is there any theoretical or empirical evidence of what happens when the propensity score estimation model is misspecified?
>
> **Response to Question 1:**  Thank you for pointing that out! In our response to W1, we have provided additional experiments of sensitivity analysis on propensity score, which indicates that our proposed method is robust to propensity score misspecification. Please refer to it for more details.
>
> > **Q2:** How do results depend on the number of candidate estimators?
>
> **Response to Q2:** Thanks for that interesting question. We provide additional experiments here under the same setting as Table 1 on IHDP. The candidate estimator list is: TARNet, Causal Forest, X-learner, S-learner. It can be seen that increasing the number of candidate estimators does not necessarily improve estimation accuracy, possibly because some of the pairs perform poorly and dilute the overall effect. However, they all outperform the baseline TARNet.
> | Number of candidate estimators | $\sqrt{\epsilon_{\text{PEHE}}^{\text{in}}}$ | $\epsilon_{\text{ATE}}^{\text{in}}$  | $\sqrt{\epsilon_{\text{PEHE}}^{\text{out}}}$| $\epsilon_{\text{ATE}}^{\text{out}}$ |
> | ---- | --- | --- | --- | --- |
> | 2 |  0.708  | 0.107  | 0.740  | 0.112 |
> | 3 |  0.638  | 0.090  | 0.670  | 0.105  |
> | 4 | 0.670  |  0.111 |  0.698 | 0.123  |
> | Baseline : TARNet | 0.896 | 0.279 | 0.920 | 0.266|
>
> ---
>
> **Reference:**
>
> [1] Imai, K. and Ratkovic, M. Covariate balancing propensity score. Journal of the Royal Statistical Society (Series B), 2014.
>
> [2] Sant'Anna, P. H. C., Song, X., and Xu, Q. Covariate distribution balance via propensity scores. Journal of Applied Econometrics, 2022.
>
> [3] Svetlana V. Belitser, Edwin P. Martens, Wiebe R. Pestman, Rolf H.H. Groenwold, Anthonius de Boer, Olaf H. Klungel, Measuring balance and model selection in propensity score methods, Pharmacoepidemiol Drug Saf, 2011.

---

> > ### Comment · Reviewer_eY6Y · 2025-11-28
> >
> > I would like to thank the authors for their detailed responses and for addressing my comments. I will therefore maintain my positive assessment of this paper.

---

### Official Review · Reviewer_UJPQ · 2025-11-01

**Soundness:** 3
**Presentation:** 3
**Contribution:** 3
**Rating:** 6
**Confidence:** 4

**Summary:**

This paper aims to address a very important and critical challenge of CATE estimator evaluation. The authors design a novel loss function and neural network architecture that produces a robust relative error estimate, which remains root-n-consistent.

**Strengths:**

1. The idea of solving the fundamental and practical CATE evaluation problem is interesting. Evaluating CATE is not easy unless we impose additional structural assumptions. Instead of just using a standard doubly-robust estimator, the authors first derive the specific theoretical conditions required for their estimator to be robust to outcome model misspecification, and then they design a new loss function and a neural network architecture to force the nuisance parameter estimates to satisfy these conditions. This is, somehow, follows the idea of the design of Dragonnet but gives an interesting solution for CATE evaluation.

2. Theoretical results are solid. They establish a theoretical asymptotic property for their proposed estimator, under some assumptions (conditions).

3. The empirical evaluation is through. The experiments strongly support the paper's claims, showing that the proposed method achieves the target 90% confidence interval coverage and high "selection accuracy"

**Weaknesses:**

1. Can strengthen the connection with other studies (see below question 2&3).

2. The asymptotic property relies on some critical assumptions, for example, $\check{\gamma}, \check{\beta}_0, \check{\beta}_1$ should converge to the true one at a rate faster than $n^{-1/4}$. It is strong but also reasonable. It is reasonable because orthogonal ML literature always assumes the convergence rate of nuisance parameters. It is strong because we are treating them as "plug-in" quantities, instead of the nuisance parameters that the estimator should be doubly robust to. It might be useful to provide some explanations to justify that the condition is realistic.

Overall, I think these are minor weaknesses. The pros outweigh the cons.

**Questions:**

1. What does the robust evaluation exactly mean? After reading this paper, I guess the author wants to claim the relative error estimator is "doubly robust", so it is robust to nuisance parameter estimation.

2. What's the difference between your loss and R-loss (R-loss is also doubly robust)?

3. The problem of robust evaluation of the CATE estimator has been studied in [1], where they also consider the worst-case performance of the CATE estimator selected by the proposed evaluation metric. Does this paper also provide any information on the worst-case performance? I guess Figure 1 provides such information, and I suggest emphasizing this point as it can strengthen the efficacy of "robust".

4. A brainstorm: I think the whole framework can be extended to policy/estimator adaptation, e.g., following the setting in [2]. Maybe we can design a new policy evaluation metric that is doubly robust to nuisance parameters, which is very useful when distribution shift presents, and it has not been discussed in previous literature. So this is also a good point of this paper, as it has the potential to be extended to other problems.

[1] Unveiling the Potential of Robustness in Selecting Conditional Average Treatment Effect Estimators

[2] Optimal Policy Adaptation under Covariate Shift

---

> ### Author Response · Authors · 2025-11-19
> **Rebuttal by Authors (1/2)**
>
> **Thank you very much for your positive evaluation of our paper. According to your comments, we have revised the manuscript carefully and thoroughly, with the major changes marked in red. Below, we hope that our clarification addresses your concerns.**
>
> >**W2:** The asymptotic property relies on some critical assumptions, for example,  $\check \gamma, \check \beta_0,  \check \beta_1$ should converge to the true one at a rate faster than $n^{-1/4}$. It is strong but also reasonable. It is reasonable because orthogonal ML literature always assumes the convergence rate of nuisance parameters. It is strong because we are treating them as "plug-in" quantities, instead of the nuisance parameters that the estimator should be doubly robust to. It might be useful to provide some explanations to justify that the condition is realistic.
>
> **Response to W2:** Thank you for raising that concern. **We would like to clarify that the convergence rates required for $\check \gamma, \check \beta_0,  \check \beta_1$ are mild.** Here are the reasons:
> - In Theorem 1, we require that "$\check \gamma, \check \beta_0,$ and  $\check \beta_1$ converge to **their probability limits** at a rate faster than $n^{-1/4}$."
>   - **The key point is that these limits are their probability limits, not the true values.**
>   - This distinction is conceptually fundamental. As explained in lines 210–212 and 326–328 of the manuscript, **$\check{\gamma}$, $\check{\beta}_0$, and $\check{\beta}_1$ always converge to their respective probability limits, regardless of whether the models are correctly specified.**
> - **In addition, the proposed method is not merely doubly robust but in fact beyond (or superior to) standard double robustness.**
>   - As shown in Theorem 1 (or lines 322–328), our estimator attains $\sqrt{n}$-consistency as long as the propensity score model is correctly specified, even if the outcome regression model is misspecified (thus beyond double robustness).
>   - In contrast, previous doubly robust methods achieve $\sqrt{n}$-consistency only when both the propensity score model and the outcome regression model are correctly specified, and when $\check{\gamma}$, $\check{\beta}_0$, and $\check{\beta}_1$ converge to **their true values** at a rate faster than $n^{-1/4}$ (see lines 167–170 for details).
>
> > **Q1:** What does the robust evaluation exactly mean? After reading this paper, I guess the author wants to claim the relative error estimator is "doubly robust", so it is robust to nuisance parameter estimation.
>
> **Response to Q1:** Thank you for your comments. We would like to clarify that our goal is not to construct a "doubly robust" estimator of the relative error. Instead, **we aim to develop an estimator of the relative error that extends beyond and improves upon standard doubly robust estimators**, thereby providing a more robust approach for evaluating CATE estimators. Here are two key points:
>
> * **Motivation: The standard doubly robust estimator is not robust enough in practice.** Specifically, achieving $\sqrt{n}$-consistency (which is needed to obtain valid confidence intervals or conduct inference) requires all nuisance estimators (the propensity score and outcome regression functions) to be consistent. This requirement can be overly stringent in real-world applications because the outcome regression model heavily relies on extrapolation and is therefore likely to yield biased estimates.
>
> * **Contribution:** We propose a robust approach for estimating the relative error. The proposed estimator retains the desirable properties of standard doubly robust estimators while **allowing for misspecification in the outcome regression functions.** As shown in Theorem 1, consistency and asymptotic normality hold as long as the propensity score is consistently estimated at a mild rate. Therefore, our method retains the desirable stability properties of DR estimators, while relaxing one of their key assumptions—outcome regression consistency—which we believe is especially valuable in CATE comparison
>
>
> > **Q2:** What's the difference between your loss and R-loss (R-loss is also doubly robust)?
>
> **Response to Q2:** Thank you for your insightful comments. We are not entirely sure what "R-loss" refers to. We assume the reviewer is asking about the differences between our proposed method and the standard doubly robust approach. If so, please kindly refer to our responses to W2 and Q1.

---

> ### Author Response · Authors · 2025-11-19
> **Rebuttal by Authors (2/2)**
>
> > **Q3:** The problem of robust evaluation of the CATE estimator has been studied in [1], where they also consider the worst-case performance of the CATE estimator selected by the proposed evaluation metric. Does this paper also provide any information on the worst-case performance? I guess Figure 1 provides such information, and I suggest emphasizing this point as it can strengthen the efficacy of "robust".
>
> **Response to Q3:** Thank you for pointing out this related literature. Although this work [1] proposes a robust evaluation method for CATE estimator by accounting for the worst-case risk, **our work is fundamentally different from it.**
> - The work[1] focuses on **distributional robustness**, evaluating CATE estimators under the worst-case counterfactual distribution within an ambiguity set that **reflects violations of unconfoundedness.**
> - Our paper, on the contrary, concerns **nuisance parameters misspecification**: we show that the proposed relative-error estimator remains the desired statistical properties even when the outcome regression functions are misspecified.
>
> The two types of robustness address fundamentally different sources of uncertainty. Nevertheless, **this work[1] offers an insightful and promising direction for potential improvement, which we have emphasized in the revised version of the manuscript.**
>
> For the comments on Figure 1, it does not provide worst-case performance; it evaluates the accuracy of the constructed confidence intervals (i.e., coverage rate) for the estimated relative error.
>
>
> > **Q4:** A brainstorm: I think the whole framework can be extended to policy/estimator adaptation, e.g., following the setting in [2]. Maybe we can design a new policy evaluation metric that is doubly robust to nuisance parameters, which is very useful when distribution shift presents, and it has not been discussed in previous literature. So this is also a good point of this paper, as it has the potential to be extended to other problems.
>
> **Response to Q4:** Thank you for raising this interesting question and for your helpful suggestions.
>
> - For clarity, we first recall our setting (see lines 107-110): Suppose we have a set of candidate CATE estimators trained on a training set,  denoted by $\lbrace \hat\tau_1(x), \cdots, \hat\tau_K(x) \rbrace$. We aim to select the estimator with the highest accuracy of CATE using **a test set $D_1 =\lbrace (X_i, A_i, Y_i): i = 1, ..., n \rbrace$.**
>
> - As suggested by the reviewer, and following the setting in [2], **suppose we have access to an additional test set $D_2 = \lbrace X_j: j =n+1, ..., n+m \rbrace $ that contains only the covariates.** If the population of interest is $D_2$, we may wish to select the optimal CATE estimator from $\lbrace\hat\tau_1(x), \cdots, \hat\tau_K(x)\rbrace$, by combining information from both $D_1$ and $D_2$. In this case, we would need to consider not only the robust properties of the relative error estimator, but also account for the covariate shift between $D_1$ and $D_2$. It is an interesting and promising direction for further exploration. This is an interesting and promising direction for future exploration. Thank you again.
>
> ---
>
> **References**
>
> [1] Unveiling the Potential of Robustness in Selecting Conditional Average Treatment Effect Estimators
>
> [2] Optimal Policy Adaptation under Covariate Shift

---

### Comment · Reviewer_M3dQ · 2025-11-20
**No revised manuscript**

The authors state that the revised manuscript includes “major changes marked in red.” However, no updated PDF reflecting these changes has been provided. The current submission appears identical to the previous version, and the claimed revisions cannot be verified.

---

> ### Author Response · Authors · 2025-11-20
> **We have uploaded our revised manuscript includes “major changes marked in red” before your message ; )**
>
> Thank you for your quick reply! We double-checked that we have successfully uploaded our revised manuscript includes “major changes marked in red” hours ago before your posting this message - please kindly refer the current PDF to evaluate our revisions. Many thanks for your time and efforts ; )

---

> > ### Comment · Reviewer_M3dQ · 2025-11-20
> >
> > Last time I checked the PDF wasn't there -- but it works now. -- Reviewer

---

> ### Author Response · Authors · 2025-11-20
> **Happy to hear that it works now**
>
> Dear Reviewer, it's good to hear that! Please let us know if we have properly addressed your questions and we are more than happy to discuss more. Again, thank you for your valuable suggestions which have undoubtedly contributed to improving the quality of our paper.

---

### Author Response · Authors · 2025-11-23
**Authors-Reviewers Discussion and Manuscript Revision Summary**

Dear reviewers and AC,

We sincerely thank all reviewers and AC for their great effort and constructive comments on our manuscript. During the rebuttal period, we have been focusing on these beneficial suggestions from the reviewers and doing our best to add several experiments and revise our manuscript. We believe our current carefully revised manuscript can address all the reviewers’ concerns.

As reviewers highlighted, we believe our paper **provides an interesting and effective solution for the evaluation of HTE estimators** (Reviewer UJPQ, Reviewer eY6Y, Reviewer GfzX). We also appreciate that the reviewers found  **the theoretical contributions strong and rigorous, especially the relaxation of key assumptions in prior work** (Reviewer UJPQ, Reviewer eY6Y, Reviewer M3dQ, Reviewer GfzX), **the proposed loss function and network architecture well-motivated and tightly connected to the theory** (Reviewer UJPQ, Reviewer M3dQ, Reviewer GfzX), as well as the **thorough empirical evaluation across multiple benchmark datasets with comprehensive ablation and sensitivity analyses** (Reviewer UJPQ, Reviewer eY6Y, Reviewer GfzX).

Moreover, we thank the reviewers for pointing out the limitations regarding **the clarity of several assumptions and modeling choices** (Reviewer UJPQ, Reviewer eY6Y, and Reviewer M3dQ), as well as for the suggestions on **strengthening the connection to related work** (Reviewer UJPQ and Reviewer M3dQ), **expanding empirical evaluations including sensitivity checks and comparisons** (Reviewer eY6Y, Reviewer M3dQ, and Reviewer GfzX), **improving consistency in terminology and presentation** (Reviewer UJPQ and Reviewer GfzX), and addressing additional concerns such as **representation design**, **convergence diagnostics**, **experimental details demonstration**, and **typos** (Reviewer M3dQ). In response to these comments, we have carefully revised and enhanced our manuscript with the following important changes with the added experiments:
- [Reviewer UJPQ, Reviewer eY6Y, and Reviewer M3dQ] We have **rewritten the assumption statements and their justifications, adding clearer explanatory discussion and explicitly emphasizing the relaxation we introduce compared with Gao (2025)** in Section 3 and Section 4.4.
- [Reviewer UJPQ and Reviewer M3dQ] We **revise and explicitly articulate how our method differs from Gao (2025) in both theory and implementation**, especially the relaxation of nuisance requirements. We have also **added Huang et al.'s work(2024) to the Conclusion section** as a possible future direction.
- [Reviewer eY6Y, and Reviewer M3dQ] We have **added a sensitivity analysis of the propensity score** to Section 5.
- [Reviewer eY6Y, and Reviewer M3dQ] We have **reorganized the experimental section** by **moving the comparison with Gao (2025) from the appendix to Section 5**, and we have **highlighted the components of the ablation study that directly contrast with Gao’s method**.
- [Reviewer M3dQ] We have **added a detailed description of the shared representation** in Section 4 and **provided an ablation study** in the appendix to illustrate its empirical contribution.
- [Reviewer M3dQ] We have **added a time–complexity analysis of the proposed HTE estimator** and reported empirical runtimes by varying both the number of candidate estimators and the sample size.
- [Reviewer eY6Y] We have **added an experiment in the appendix that examines how varying the number of candidate HTE estimators affects the prediction results**.
- [Reviewer M3dQ] We have **rewritten the paragraph introducing $L_{\text{const}}$ and $L_{\text{ce}}$**, and **added an experiment showing empirically that, in finite samples, the two losses effectively approximate the original conditions** in the appendix **(new edit at 11/23)**.
- [Reviewer M3dQ] We have **added a sensitivity analysis of the shared representation** in the appendix.
- [Reviewer M3dQ] We have **added additional experiments on ACIC 2016 and News** in the appendix  **(new edit at 12/01)**.
- [Reviewer eY6Y, Reviewer M3dQ, and Reviewer GfzX] We have **corrected typos**, **harmonized terminology** throughout the paper, and **reorganized several sections** to improve structural coherence. We have also **corrected several language issues**  **(new edit at 12/01)**.

Revisions are temporarily highlighted for clarity: changes made earlier are shown in $\textcolor{red}{red}$, and the new updates made on Dec 1 are shown in $\textcolor{blue}{blue}$.

We hope this summary sufficiently captures the author–reviewer discussion and clearly addresses all of the reviewers’ concerns.

Many thanks,

Submission24523 Authors

---

### Meta-Review · Area_Chair_ZHnj · 2026-01-06

**Summary:**

The paper proposes a robust evaluation framework for Heterogeneous Treatment Effect (HTE) estimators based on relative error. The core contribution is a method to estimate the relative performance difference between two HTE estimators even when nuisance parameters (specifically the outcome regression model) are misspecified. The authors derive theoretical conditions for robustness, propose a specific loss function and neural network architecture to satisfy these conditions, and introduce an enhanced HTE estimator as a byproduct.

**Reviewer Concerns:**

There is a clear split among the reviewers. Three reviewers (UJPQ, eY6Y, GfzX) are strongly positive (scores 6, 8, 8), praising the theoretical contribution of relaxing consistency assumptions, the novelty of the evaluation framework, and the thorough empirical validation. One reviewer (M3dQ) is negative (score 2), criticizing the "heuristic" nature of the optimization, the feasibility of the moment conditions, and the practical relevance given that propensity scores are rarely correctly specified.

Meta-Reviewer Judgment
The paper makes a significant theoretical and practical contribution to the evaluation of HTE estimators, a critical but under-studied area. The proposed framework offers a principled way to compare estimators with weaker assumptions than prior work (specifically Gao 2025).

Regarding the negative reviewer (M3dQ)'s concerns:
Reliability/Infeasibility: The reviewer argues that the method relies on moment conditions that are "infeasible" in finite samples. The authors countered effectively by explaining that these are standard balancing conditions used widely in causal inference (e.g., CBPS) and provided empirical evidence that their soft-penalty approach satisfies these constraints well in practice.

Propensity Score Misspecification: The reviewer is concerned that propensity scores are rarely correct. The authors showed via sensitivity analysis that the method is robust to reasonable noise in the propensity score. Furthermore, the assumption of a correctly specified propensity model is standard in this literature to achieve identification/robustness properties, and the authors' relaxation of the outcome model specification is a genuine advancement.

Enhanced Estimator: The reviewer critiqued the enhanced estimator for modeling full potential outcomes. The authors clarified this is an ancillary contribution (a byproduct of the evaluation framework) and does not detract from the main contribution.

The positive reviewers highlighted the value of the problem, the rigor of the relaxation of assumptions, and the effectiveness of the proposed solution. The authors' rebuttal was comprehensive, adding experiments on new datasets (ACIC, News) and clarifications on theory that satisfied the majority.

**Reviewer Scores:**

Reviewer UJPQ (Score: 6; likely no change): The authors provided detailed clarifications regarding the theoretical assumptions and the distinction of their method from standard doubly robust approaches. As the reviewer’s initial assessment was positive and the questions were addressed without follow-up objections, it is likely their score would have remained at 6.

Reviewer eY6Y (Score: 8; likely no change): This reviewer explicitly confirmed in the discussion that they would maintain their positive assessment (Score: 8) following the authors' sensitivity analyses and clarifications regarding propensity score misspecification.

Reviewer M3dQ (Score; likely no change): Despite the authors adding significant new experiments (ACIC, News datasets) and sensitivity analyses in response to this reviewer, the reviewer remained unconvinced about the fundamental framing ("reliability") and the practical feasibility of the moment conditions. While they acknowledged the validity of the optimization experiments, they maintained that the paper was "not yet ready." It is unlikely their score would have risen above a 3.

Reviewer GfzX (Score: 8; likely no change): This reviewer explicitly stated in the discussion that they remain positive and are keeping their original overall score of 8, while raising their presentation score.

---

### Decision · Program_Chairs · 2026-01-26

Accept (Poster)